# Learning Tree-Structured Composition of Data Augmentation

**Dongyue Li** *li.dongyu@northeastern.edu*
*Northeastern University, Boston*

**Kailai Chen** *chen.kailai@northeastern.edu*
*Northeastern University, Boston*

**Predrag Radivojac** *predrag@northeastern.edu*
*Northeastern University, Boston*

**Hongyang R. Zhang** *ho.zhang@northeastern.edu*
*Northeastern University, Boston*

**Reviewed on OpenReview:** *https://openreview.net/forum?id=lmgf03HeqV*

## Abstract

Data augmentation is widely used in scenarios where one needs to train a neural network given little labeled data. A common practice of augmentation training is applying a composition of multiple transformations sequentially to the data. Existing augmentation methods such as RandAugment rely on domain expertise to select a list of transformations, while other methods such as AutoAugment formulate an optimization problem over a search space of size $k^d$, which is the number of sequences of length $d$, given a list of $k$ transformation functions.

In this paper, we focus on designing efficient algorithms whose running time complexity is much faster than the worst-case complexity of $O(k^d)$, provably. We propose a new algorithm to search for a binary tree-structured composition of $k$ transformations, where each tree node corresponds to one transformation. The binary tree generalizes sequential augmentations, such as the one constructed by SimCLR. Using a top-down, recursive search procedure, our algorithm achieves a runtime complexity of $O(2^d k)$, which is much faster than $O(k^d)$ as $k$ increases above 2. We apply the algorithm to tackle data distributions with heterogeneous subpopulations, by searching for one tree in each subpopulation, and then learn a weighted combination, leading to a *forest* of the trees.

We validate the proposed algorithms on numerous graph and image data sets, including a multi-label graph classification data set we collected. The data set exhibits significant variations in the sizes of graphs and their average degrees, making it ideal for studying data augmentation. We show that our approach can reduce the computation cost (measured by GPU hours) by 43% over existing augmentation search methods while improving performance by 4.3%. Extensive experiments on contrastive learning also validate the benefit of our approach. The tree structures can be used to interpret the relative importance of each transformation, such as identifying the important transformations on small vs. large graphs.

## 1 Introduction

Data augmentation is a technique to expand the training data set of a machine learning algorithm by transforming data samples with pre-defined transformation functions. Data augmentation is widely used in settings where only a small fraction of the unlabeled data is annotated, such as self-supervised learning (Xie et al., 2020; Chen et al., 2020b). A common practice of augmentation training is to apply a sequence of transformations (Ratner et al., 2017; Cubuk et al., 2019; Lim et al., 2019; Zhang et al., 2020). For instance, AutoAugment (Cubuk et al., 2019) first applies random rotation to the images, then equalizes the histogram

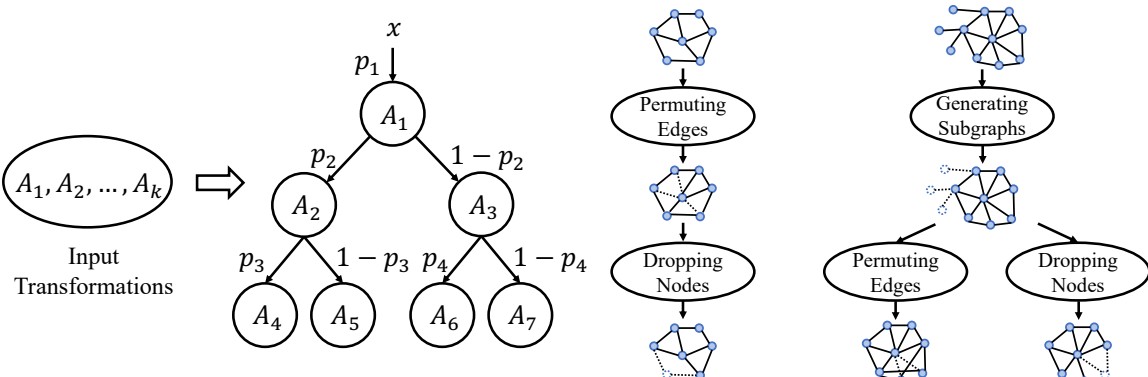

Figure 1: We illustrate the overall procedure of our algorithms. The input consists of $k$ transformation functions, denoted as $A_1, \ldots, A_k$. Given a data set, our algorithm constructs a probablistic binary tree-structured composition of these transformations, as shown on the left. Given an input data $x$, $A_1$ is applied to map $x$ to $A_1(x)$, with probability $p_1$; otherwise, no transformation is applied, and $x$ remains unchanged. Let $x'$ denote the output. In the next step, we will apply $A_2$ to $x'$ with probability $p_2$, or $A_3$ to $x'$ with probability $1 - p_2$, etc. The second algorithm will first partition the entire data set into a few groups, e.g., by the sizes of graphs, as illustrated above. Then, the first algorithm is applied to learn one tree for each partition. These trees are weighted jointly to form a "forest" as the final augmentation scheme. A byproduct is that we can now measure the importance of each transformation in the tree of each group. For example, we find that permuting edges by randomly adding or deleting a fraction of edges works best for small graphs. For large graphs, generating a subgraph by simulating a random walk works better. Notice that in an augmentation tree, if a branch only has a single child node, it means if the transformation is not applied, then we will not change the input or use any augmentation, which is the same as applying $A(x) = x$.

of each cell. The sequence of transformations to apply depends on the downstream application; For instance, transformations that apply to colored images may not be suitable for black-and-white images. Thus, it would be desirable to have a procedure that, given a list of transformations and a target data set of interest, finds the most beneficial composition of transformations for that data set.

Various techniques have been introduced in prior work to overcome the computational challenge of finding transformation compositions, which often involve solving a highly complex optimization problem with a worst-case complexity of $O(k^d)$ (Ratner et al., 2017; Cubuk et al., 2019). Concretely, if there are $k$ transformations, and the goal is to find the best sequence of length $d$, the search space would have $k^d$ possible choices. The abovementioned optimization methods suffer from a worst-case complexity of $O(k^d)$ because the search space includes all the $k^d$ choices. This paper aims to revisit this issue in a range of settings of strong practical interest by designing search algorithms whose runtime complexity is much faster than this worst-case.

As an example, we start by considering a data set of proteins with the goal of predicting their biological activity (function), a task relevant to biological discovery (Radivojac, 2022) and precision medicine (Rost et al., 2016). Each protein is given as a graph, with nodes corresponding to amino acid residues and edges indicating spatial proximity between the residues in a protein's 3D structure. Protein function prediction can be seen here as a graph-level multi-label classification problem (Clark & Radivojac, 2011), where the presence of a specific function indicates a binary label and proteins can have multiple functions. Since the graphs can vary considerably in size and degree distribution, finding the right augmentation for this type of data set is particularly challenging. On a related note, see also Zhang et al. (2022); Gao et al. (2023); Nguyen et al. (2023) for studies on image data sets with stratified subpopulations.

Existing approaches for finding compositionality either rely on domain expertise to identify the most relevant transformations, or tackle a hyper-parameter optimization problem over the search space (Wu et al., 2020). For instance, reinforcement learning-based methods are one type of approach (Cubuk et al., 2019; Luo et al., 2023), which define a reward function given an augmentation, and then search over the space of compositions

to optimize the reward function. These methods require exploring a search space of size $k^d$ for finding a sequence of length $d$. Therefore, in the worst case, the runtime complexity can still be $O(k^d)$.

The contribution of this paper is designing a faster algorithm to find a binary tree composition of $k$ transformations of depth $d$, whose running time complexity is instead only $O(2^d k)$, provably. Our empirical results show that this complexity reduction does not come at the cost of downstream performance. The algorithm conducts a top-down, recursive search to construct a binary tree, where each node of the tree corresponds to one transformation. This tree structure leads to a natural notion of importance score for each transformation, analogous to the feature importance score used in tree-based methods. Additionally, the algorithm can be used to tackle the abovementioned scenario where the underlying data involves a mixture of stratified subpopulations. See Figure 1 for an overall illustration of our approach.

We conduct extensive experiments on numerous data sets to validate our proposed algorithms. First, we apply our algorithm to a newly collected graph classification data set generated using AlphaFold2 protein structure prediction APIs (Jumper et al., 2021), containing 20,504 human proteins and 1,198 types of protein functions listed in Gene Ontology (Ashburner et al., 2000). Compared to existing data sets in Borgwardt et al. (2005) and Hu et al. (2020a), our data set is based on a newer source and contains a broader set of protein functions. We find that our algorithm can outperform RandAugment which does not search for composition by **4.3**%. Compared to recent augmentation optimization methods such as GraphAug (Luo et al., 2023), our algorithm reduces its runtime by **43**% and achieves **1.9**% better performance. Second, we evaluate our algorithm in contrastive learning settings. Compared to the augmentation method of Chen et al. (2020b), the performance is on par with natural images such as CIFAR-10. On a medical image classification data set, our algorithm can find a better augmentation scheme that outperforms SimCLR by **5.9**%. Finally, we justify the design of our algorithm through a detailed ablation analysis. The details can be found in the experiments section.

To summarize, we list the main results of our paper as follows:

- We design an algorithm whose worst-case running time complexity is $O(2^d k)$, where $k$ is the number of input transformations and $d$ is the length of the composition. This is a significant improvement compared to the worst-case complexity of $O(k^d)$ (e.g., when $k > 10$ and $d > 3$).

- We use the algorithm to tackle data distributions that involve a mixture of heterogeneous subpopulations, by first partitioning the data set into several groups, e.g., according to the graph sizes. The algorithm is tested on a newly collected graph classification dataset and a few other benchmark data sets, reducing runtime while showing improved performance compared to several search methods.

- We construct a new multi-label graph classification data set, which contains a wide spectrum of graph sizes and degrees, making it ideal for studying data augmentation in future work.

## 1.1 Related Work

Motivated by the computational considerations around AutoAugment, active sampling has been used to accelerate the search procedure in data augmentation training (Wu et al., 2020). Besides, bilevel optimization (Benton et al., 2020), which applies a weighted training procedure on the augmentations, is another effective approach. Yang et al. (2022) introduce an adversarial training objective to find hard positive examples. Hounie et al. (2023) formulate data augmentation as an invariance-constrained learning problem and leverage Monte Carlo Markov Chain (MCMC) sampling to solve it. It is worth noting that the tree structures we are proposing in this work can be adopted as the base compositionality in all of these optimization frameworks. Further exploring the utility of tree structures would be an interesting direction for future work.

Apart from supervised learning, data augmentation plays a crucial role in contrastive learning (You et al., 2020). The invariance introduced by data transformation methods can serve as a regularizer in self and semi-supervised learning techniques (Xie et al., 2020). Related to graph structures, there is a line of work on model robustness to graph size shifts by comparing local graph structures (Yehudai et al., 2021), or by applying regularization on representation distances between different sizes (Buffelli et al., 2022) (see the discussion in Ju et al. (2023) for further references). In particular, Li et al. (2023) and Nippani et al. (2023) design multitask learning algorithms to improve the robustness of learning under distributional shifts or data

imbalance. Our work contributes to this line of literature by developing a simple, efficient data augmentation search procedure. Besides, there has been developments on novel use cases of higher-order graph structures in graph neural networks (Chien et al., 2022), and spectral clustering within a Superimposed Stochastic Block Model (Paul et al., 2023). Specific to graph contrastive learning, little is known regarding the use of higher-order graph structures. Further fleshing this out may be a promising avenue for future work.

Even though data augmentation has generally been quite useful in practice, the theoretical understanding of data augmentation training is relatively scarce. Part of the challenge is that the data augmentation training paradigm violates the independent sampling assumption typically required by supervised learning. For instance, suppose one would like to understand how a sequence of transformations, like rotation, cropping, etc., to an image, affects the downstream performance. This would require modeling such transformations within the data augmentation paradigm. There have been few developments in this direction. Dao et al. (2019) model the behavior of stochastic augmentation in a Markov chain, and reason about its behavior more precisely using kernel methods. Chen et al. (2020a) develop a group-theoretical framework modeling data augmentation. Their framework applies to the family of label-preserving transformations. Wu et al. (2020) categorize the generalization effects of linear transformations using a bias-variance decomposition and study their effects on the ridge estimator in an over-parameterized linear regression setting. Shao et al. (2022) develop a PAC-learning framework for transformation-variant hypothesis spaces.

Specific to the approach taken by this paper, we note that greedy recursive partitioning is a known method for constructing decision trees. There are studies, albeit in a very different setting, on the approximation ratio of the greedy search for constructing trees (e.g., Adler & Heeringa (2008); Gupta et al. (2017)). It is an interesting question to revisit these results in the setting of designing tree data augmentation.

**Organization.** We start with our problem setup in Section 2. Then, we describe the proposed algorithms in Section 3. We will also give some examples of the algorithms to illustrate their design. Next, we present the empirical evaluations in Section 4, with a detailed comparison to existing approaches. The Appendix includes theoretical derivations and analysis omitted from the main text.

## 2 Learning Composition of Data Augmentation

This section describes the problem setup for learning a composition of transformations. Consider a prediction problem where the goal is to map an input feature vector $x \in \mathcal{X}$ to a label $y \in \mathcal{Y}$. We have access to a data set $\hat{P}$, which includes a list of examples $(x, y)$ drawn independently from an unknown distribution $\mathcal{P}$ over $\mathcal{X} \times \mathcal{Y}$. Given $k$ transformation functions, denoted as $A_1, \ldots, A_k$, each transforms an input feature vector from $\mathcal{X} \to \mathcal{X}$. The problem is to find a composition $Q$ of a subset of the $k$ functions, such that a model $f_\theta$ is learned to minimize the loss, denoted as $\ell$, of the augmented examples. For example, in the case that $\ell$ is the cross-entropy loss over $C$ classes, it maps from $\mathbb{R}^C \times \mathcal{Y} \to \mathbb{R}$. We will design $Q$ with a probabilistic distribution over sequences of compositions. Concretely, let $\tau$ be a sequence sampled from $Q$. The learning objective can be written as:

$$L(f_\theta; Q) = \mathop{\mathbb{E}}_{(x,y) \sim \mathcal{P}} \mathop{\mathbb{E}}_{\tau \sim Q} [\ell(f_\theta(\tau(x)), y)].$$

When the underlying distribution involves several groups of stratified subpopulations, we can write the above objective as a mixture of losses over the groups. Let $m$ denote the number of groups and let $\mathcal{G} = \{1, \ldots, m\}$ denote the set of group labels. Let $P_g$ denote the data distribution of $\mathcal{P}$ restricted to group $g$. For our context, it is sufficient to think that the group labels are available during training and testing time. For this setting, the learning objective becomes

$$L(f_\theta; Q) = \sum_{g=1}^{m} q_g \cdot L_g(f_\theta; Q), \text{ where } L_g(f_\theta; Q) = \mathop{\mathbb{E}}_{(x,y) \sim P_g} \mathop{\mathbb{E}}_{\tau \sim Q} [\ell(f_\theta(\tau(x), y)], \tag{1}$$

and $q_g \in (0, 1)$ indicates the proportion of examples coming from group $g$. Besides this empirical risk minimization setup, an alternative min-max optimization is studied in robust optimization (Zhang et al., 2022; Nguyen et al., 2023). We note that the ideas to be presented next can also be used in this min-max formulation.

One type of compositionality is using a probabilistic sequence. We illustrate such an example in Figure 2, used in SimCLR (Chen et al., 2020b). This example applies "random cropping" with probability $p$. With probability $1 - p$, the input is not changed, and is pushed to "color distortion," and finally "Gaussian blurring." The probability value of each transformation indicates the likelihood of applying it at that step. More generally, consider a sequence of length $d$, denoted as $(A_1, p_1), (A_2, p_2), \ldots, (A_d, p_d)$, where each $A_i$ is associated with $p_i \in [0, 1]$. The transformations are applied sequentially from the first node to the last node, each in a probabilistic way. For example, if $A_i$ is not applied, then the input to $A_i$ remains unchanged and is passed to the next node, similar to a skip connection, and so on.

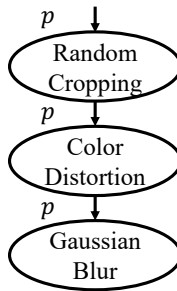

Figure 2: Illustrating a sequential augmentation scheme (Chen et al., 2020b).

In this paper, we will consider the family of binary tree-structured compositions, which naturally generalizes the sequence illustrated in Figure 2. We define the compositionality as a binary tree. With depth $d$, there will be $2^d - 1$ nodes in total. Each node is associated with a transformation $A_i$ and the probability value $p_i$, indexed from $i = 1, \ldots, 2^d - 1$. Given such a tree, we apply the transformations from the root node to one of the leaf nodes. For instance, after applying $A_i$, the augmentation proceeds to either the left or right node. A special case is when the node itself is the identity mapping. In such cases, the transformation will terminate at that point.

We now elaborate on the computational cost of existing methods. AutoAugment (Cubuk et al., 2019) uses reinforcement learning as the search method to find a composition that optimizes the validation performance. This method takes many GPU hours to search for a composition with depth $d = 2$ out of $k = 16$ augmentations on CIFAR-10. Fast AutoAugment (Lim et al., 2019) uses Bayesian optimization as the search method. Both of these two methods use a worst-case search space of $O(k^d)$. Instead of searching in a discrete space, Chatzipantazis et al. (2023) parameterizes the composition space with $O(k^d)$ continuous variables, and then uses SGD to optimize the model weights and these variables jointly.

## 3    Our Proposed Algorithms

We now present our algorithm for learning tree compositions. Our algorithm conducts a greedy search from the root to one of the leaves. We will provide illustrative examples to validate this algorithm. Then, we extend the algorithm to a setting involving multiple subpopulations. We will design a weighted, probabilistic combination of the trees in a forest.

### 3.1    Learning Tree-Structured Composition

We consider a top-down binary search procedure inspired by the *recursive binary splitting* of building a decision tree. The procedure searches from the root of a tree and iteratively finds one augmentation in one of the empty leaf nodes. A node would indicate a new branch of two nodes to build the tree further.

For a particular node $i$ in the tree, we search for the choice of $A_i$ and $p_i$ that leads to the best improvement in the validation performance, such as cross-entropy loss. Each search iterates over all possible transformations $A_1, \ldots, A_k$ and a discrete list of probabilities. We include $A(x) = x$, namely identity mapping, to one of the $k$ transformations. Naively optimizing the validation performance requires training $O(k)$ models for each choice and choosing one via cross-validation. This can be slow when $k$ is large.

We enhance the efficiency of each search step by employing a density matching technique, without repeatedly training multiple models. This method has been used in prior work (Lim et al., 2019). Specifically, we train one model with the current tree, denoted as $f_{\theta^\star}$. Then, for each choice of $A$ and $p$, we incorporate them at position $i$ of the current composition and evaluate the performance of $f_{\theta^\star}$ on $n$ augmented examples generated by applying the new composition to the validation set. Denote a tree composition as $T$. We measure its

performance on a validation set of size $n$ as:

$$L_{\text{val}}(T) = \frac{1}{n} \sum_{i=1}^{n} \mathbb{E}_{\tau \sim T} [\ell(f_{\theta^\star}(\tau(x_i)), y_i)]. \tag{2}$$

Thus, each search only trains one model, with $O(k)$ evaluations of adding each transformation to position $i$ of $T$. Empirically, we find that density matching identifies the same tree as fully-trained models on a protein graph data set and another image classification dataset. We evaluate the relative residual sum of squares error between $L_{\text{val}}(T)$ and that of training a model for each transformation. We find that the gap is $\leq 0.7\%$.

This concludes the search procedure for a single tree node. We will repeat the same procedure for all the other nodes of the tree:

- When a child node is added with transformation $A_i$ and probability $p_i$, the other child node of the same parent node should find a transformation from one of the $k$ input ones, and apply the chosen transformation with probability $1 - p_i$. In other words, the probability value $1 - p_i$ will be fixed.

- The process continues until the tree reaches a pre-specified depth $d$, or until $L_{\text{val}}$ no longer improves after adding any transformation.

We summarize the complete procedure in Algorithm 1.

---

**Algorithm 1 A top-down recursive search procedure for finding a binary tree-structured data augmentation scheme**

---

**Input**: $k$ transformation functions $A_1, A_2, \ldots, A_k$ (including identity mapping), training and validation sets
**Require:** Maximum tree depth $d$; A list $H$ of probability values
**Output**: A probabilistic, binary tree-structured composition $T$ of $\{A_1, A_2, \ldots, A_k\}$

1: Initialize $T = \{\}$, $V = \{1\}$, $L_{\text{val}} = \infty$
2: **while** depth$(T) \leq d$ and $V$ is not empty **do**
3:     Randomly choose $i \in V$, let $A_i$, $p_i$, $L_{\text{val}}^{(i)} \leftarrow$ *Build-one-node*$(i, T)$, then remove $i$ from $V$
4:     Add $A_i$, $p_i$ to index $i$ of $T$     /* If $A_i$ is the identity mapping, it means no improvement is found */
5:     **if** $L_{\text{val}}^{(i)} < L_{\text{val}}$ **then**
6:         Update $L_{\text{val}} \leftarrow L_{\text{val}}^{(i)}$ and $V \leftarrow V \cup \{2i, 2i+1\}$
7:     **end if**
8: **end while**
9: **return** $T$
10: **procedure** *Build-one-node*$(i, T)$
11:     Train a model with $T$ denoted as $\theta^\star$
12:     **if** $i$'s has a sibling node that is in $T$ **then**
13:         Let $p = 1 - p_k$, where $k$ is the index of the sibling node of $i$
14:         **for** $j \in \{1, \ldots, k\}$ **do**
15:             Add $(A_j, p)$ to at position $i$ of $T$ and evaluate $L_{\text{val}}(T)$
16:         **end for**
17:     **else**
18:         **for** $j \in \{1, \ldots, k\}$ and $p \in H$ **do**
19:             Add $(A_j, p)$ to position $i$ of $T$ and evaluate $L_{\text{val}}(T)$
20:         **end for**
21:     **end if**
22:     **return** $(A_j, p)$ achieving the lowest $L_{\text{val}}$
23: **end procedure**

---

**Importance scores.** The tree structures allow us to interpret the relative importance of each transformation, analogous to feature importance scores used in tree-based methods. Concretely, we can measure the usefulness

of each transformation by the amount by which it reduces $L_{\text{val}}$, added across all of its appearances in the tree. Thus, a higher score indicates that a transformation contributes more to reducing the loss. We will refer to this as the importance score of a transformation and measure it in the experiments later on.

### 3.1.1 Running Time Analysis

We examine the running time of building a single tree. At depth $d$, the number of search steps is at most $2^d - 1$, the maximum size of the tree. Each step involves training one model and iterating over the list of $k$ transformations and probability values. Thus, the overall procedure takes $O(2^d k)$ time to complete the search. By contrast, the worst-case complexity of existing methods scales as $O(k^d)$ which is the space of all possible sequences of length $k$. We present the comparison in Table 1.

Table 1: Summary of comparisons between our approach and previous optimization algorithms to search for compositions of data augmentations. We use $d$ to denote a composition's length and $k$ to denote the number of transformations to the input.

|  | Method | Running Time | Compositionality |
|---|---|---|---|
| AutoAugment (Cubuk et al., 2019) | Reinforcement Learning | $O(k^d)$ | Sequences |
| Fast AutoAugment (Lim et al., 2019) | Bayesian Optimization | $O(k^d)$ | Sequences |
| SCALE (Chatzipantazis et al., 2023) | Stochastic Gradient Descent | $O(k^d)$ | Sequences |
| **Algorithm 1 (Ours)** | Binary Search | $\mathbf{O(2^d k)}$ | Forest of Trees |

### 3.1.2 Illustrative Examples

Next, we give an example of running our algorithm before getting further into the technical details. We will illustrate its runtime as well as previous methods's runtime. We train WideResNet-28-10 on CIFAR-10 and CIFAR-100 in supervised learning, following the setting of AutoAugment. We use $k = 16$ transformations, each with five values of perturbation scales (Cubuk et al., 2019), including ShearX/Y, TranslateX/Y, Rotate, AutoContrast, Invert, Equalize, Solarize, Posterize, Contrast, Color, Brightness, Sharpness, Cutout, and Sample Pairing. As for $H$, we set it as $0, 0.1, 0.2, \ldots, 1.0$. We use a reduced set of randomly sampled $4,000$ examples as the training set during the search. After the search, we train a model on the full data set and report the test performance. For each algorithm, we report the runtime using an Nvidia RTX 6000 GPU.

The results are shown in Table 2. Our algorithm uses 4.7 GPU hours to complete, reducing the runtime by $51\%$ compared to the existing methods. Meanwhile, the test performance remains close to the methods.

Table 2: We compare our algorithm with several existing approaches, trained from randomly initialized WideResNet-28-10 on CIFAR-10 and CIFAR-100. We report the number of GPU hours of the search on a single GPU. We also report the error rate on the test set, averaged over five random seeds.

|  | CIFAR-10 | | CIFAR-100 | |
|---|---|---|---|---|
|  | GPU hours | Error rates (%) | GPU hours | Error rates (%) |
| AutoAugment | ≥5000 | 2.6±0.1 | ≥5000 | 17.1±0.3 |
| Fast AutoAugment | 19.2 | 2.7±0.1 | 19.6 | 17.3±0.2 |
| SCALE | 9.6 | 3.3±0.1 | 9.6 | 17.3±0.1 |
| **Algorithm 1 (Ours)** | **4.7** | 3.1±0.1 | **4.7** | 17.3±0.1 |

Next, we show an example of using augmentation composition for training ResNet-50 on CIFAR-10 in contrastive learning (Chen et al., 2020b). We use seven transformations, including Random Cropping, Cutout, Rotation, Color Distortion, Sobel filtering, Gaussian noise, Gaussian blur, and the same $H$.

Figure 3 shows the tree found by our algorithm. Interestingly, this tree shares three nodes, as in SimCLR, which is carefully designed by domain experts. The tree begins by applying random cropping and color distortion, similar to SimCLR augmentation. Then, the tree uses Gaussian blur as one branch but adds rotation as another branch. As shown in Table 2, this is comparable to the tree illustrated in Figure 2.

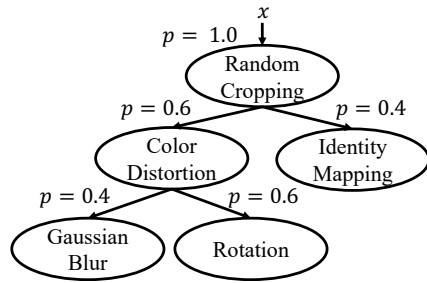

Figure 3: Illustrating the binary tree returned by Algorithm 1, conducted on CIFAR-10. On the right branch, no further transformation is applied after the identity mapping.

Lastly, we also find that the top-down search performs comparably to the exhaustive search of all possible trees of depth two. On several image and graph classification data sets, the greedy search is only 0.4% below the test result of the exhaustive search on average.

## 3.2 Learning a Forest of Trees

With the efficient search procedure, we proceed to study the case of learning from multiple subpopulations. We start by presenting a motivating example, showing that learning a single tree does not suffice. We conduct experiments on a protein graph classification data set. The problem involves multiple binary-labeled classifications, aiming to classify proteins into their protein functions, each treated as a zero-one label. Since labeling the protein functions requires domain expertise, we could only obtain about $4e^{-3}$ labels of all the unlabeled data. We will design data augmentation schemes to generate synthetic labels based on the labeled examples. The dataset exhibits significant variations in graph sizes and degrees. The sizes of the graphs range from 16 to 34,350 nodes, and the average degrees range from 1.8 to 9.4. In the experiment, we will divide the data set into 16 groups, which corresponds to 16 intervals of sizes and average degrees. We will use graph neural networks as the base learner (Hu et al., 2020b). We consider four augmentation methods, including DropNodes, PermuteEdges, Subgraph, and MaskingNodes, as in You et al. (2020). We use the same list $H$.

We first find a tree by minimizing the average loss. We also apply this to each group and compare it with a model trained without augmentation. For graphs with less than 200 nodes or an average degree of less than 0.1, just minimizing the average loss can lead to a worse result than stand-alone training.

Next, we search for one tree in each group, across all groups. For each group, we run Algorithm 1 to construct a tree. Notice that the data we use for executing this step is separate across groups. Hence, there is no synchronization performed on different groups. In other words, the models trained in each group will not be reused in training other groups (or in the weighted training step later). This leads to some heterogeneity in the tree structure we obtain for different groups. To give an example, in Figure 4, we plot two trees, corresponding to a group of the smallest sizes and another group of the largest sizes. The latter uses larger scales and one more step. We find that "Permuting Edges" and "Dropping Nodes" are used on the small-sized graphs, which remove randomly sampled edges and nodes, respectively. Meanwhile, all the transformations are used on the large graphs, including "Generating Subgraph," which extracts a subgraph by random walks, and "Masking Nodes," which randomly masks a fraction of node attributes to zero. We also report the importance score of each transformation. In Figure 4a, "Permuting Edges" has the highest importance score. Whereas in Figure 4b, "Generating subgraph" has the highest importance score.

We give another example from an image classification data set. This dataset contains three groups of images with different colors and backgrounds. We illustrate the trees found for one group of colored images and another group of black-and-white images. We note distinct differences between the two trees. Transformations that alter the color distributions, such as Auto Contrast, Solarize, and Color Enhancing, are applied to the colored images. In contrast, transformations applied to black-and-white images modify the grayscale or shape, including Equalize, TranslateY, and Posterize. Additionally, for the colored images, the Auto Contrast transformation has the highest importance score, while for the black-and-white images, Equalize has the highest importance score. These observations validate that augmentation schemes vary across groups.

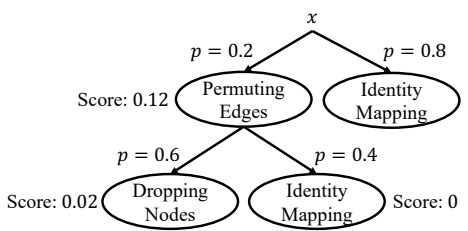 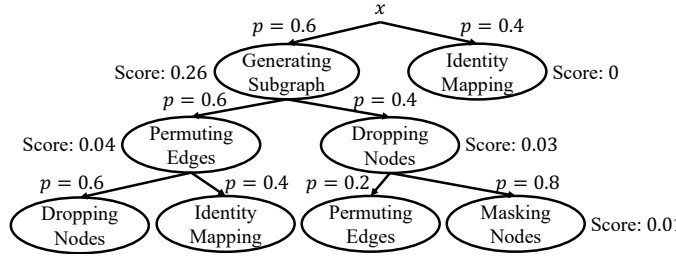

(a) Augmentation tree found on small graphs with the number of nodes less than 200.

(b) Augmentation tree found on larger graphs with the number of nodes larger than 600.

Figure 4: The augmentation trees found from different groups can vary dramatically. On a protein graph classification data set, the augmentation tree on small graphs (left) involves fewer augmentation steps than the tree found on large graphs (right). We also report each augmentation's importance score, computed from the validation set. To clarify, after the identity map, i.e., $A(x) = x$, no further transformation will be applied.

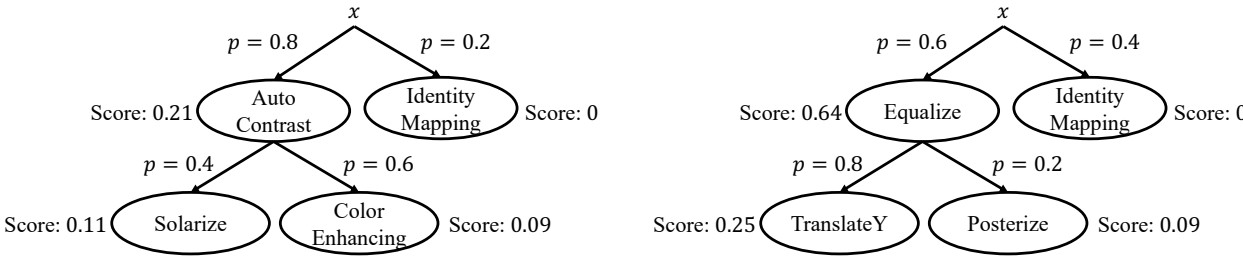

(a) Augmentation tree found on colored images.

(b) Augmentation tree found on black-white images.

Figure 5: Illustrating the trees found between colored images and black-white images on an image classification data set. The tree on the left involves different transformations compared to the right.

### 3.2.1 Learning the Weight of Each Tree

With one tree learned for each group, we next unify the trees through weighted training of a new model $f_\theta$ and the weight of each group. For each group $g$, denote the augmentation tree as $Q_g$, for $g = 1, 2, \ldots, m$. Consider a bilevel optimization problem:

$$\min_{w, \theta^w} \hat{L}(f_{\theta^w}), \quad \text{such that} \quad \theta^w \in \arg\min_\theta \sum_{g=1}^m w_g \hat{L}_g(f_\theta; Q_g). \tag{3}$$

where the loss of group $i$ is associated with a weight $w_i \in \mathbb{R}(0,1)$, and $\sum_{i=1}^m w_i = 1$. We write objective (3) as a function of $\theta^w$:

$$\min_{w \in \mathbb{R}^m} \hat{L}(f_{\theta^w}) = \sum_{g=1}^m q_g \cdot \hat{L}_g(f_{\theta^w}; Q_g). \tag{4}$$

We derive the gradient of objective (4) with respect to $w$. This is achieved by applying the chain rule and computing the gradient of $\theta^w$, for any $i = 1, \ldots, m$:

$$d_i := \frac{\partial \hat{L}(f_{\theta^w})}{\partial w_i} = -\left( \sum_{g=1}^m q_g \nabla_\theta \hat{L}_g(f_{\theta^w}; Q_g) \right)^\top H^{-1} \nabla_\theta \hat{L}_i(f_{\theta^w}; Q_i), \quad \text{where } H = \sum_{g=1}^m w_g \nabla_\theta^2 \hat{L}_g(f_{\theta^w}; Q_g). \tag{5}$$

We defer the complete derivation to Appendix A. $d_i$ can be viewed as a similarity value of group $i$ compared to the averaged group, normalized by Hessian inverse.

We will use alternating minimization to update both $\theta$ and $w$. Let the model parameters and weights at iteration $t$ be $\theta^{(t)}$ and $w^{(t)}$.

- With a fixed $w^{(t)}$, update the model parameters from $\theta^{(t)}$ to $\theta^{(t+1)}$ by running $\alpha$ SGD steps:

$$\min_{\theta} \sum_{g=1}^{m} w_g^{(t)} \cdot \hat{L}_g(f_\theta; Q_g).$$

- With a fixed $\theta^{(t+1)}$, obtain the gradient $d_i^{(t)}$ from equation (5). Then, update $w_i^{(t)}$ as:

$$w_i^{(t+1)} = \frac{w_i^{(t)} \exp\left(-\eta d_i^{(t)}\right)}{\sum_{j=1}^{m} w_j^{(t)} \exp\left(-\eta d_j^{(t)}\right)}. \tag{6}$$

Taken together, we summarize the complete procedure for learning a unified augmentation scheme in Algorithm 2. To recap, the final augmentation scheme is a probabilistic mixture of $m$ trees, one for each group.

---

**Algorithm 2 Learning a forest of trees from a mixture of subpopulations**

**Input**: $k$ transformations $A_1, A_2, \ldots, A_k$; Training/Validation splits of $m$ groups
**Require:** Number of iterations $S$, SGD steps $\alpha$, learning rate $\eta$
**Output**: $m$ trees with a probability value for each tree: $(Q_1, w_1), (Q_2, w_2), \ldots, (Q_m, w_m)$; Model weight $\theta^\star$

1: **for** $g = 1, \ldots, m$ **do**
2:     Compute an augmentation scheme $Q_g$ for group $g$ using Algorithm 1
3: **end for**
4: Initialize model parameters $\theta^{(0)}$; Set weight variables $w^{(0)}$ as the uniform proportions $[1/m, \ldots, 1/m]$
5: **for** $t = 0, \ldots, S - 1$ **do**
6:     Update $\theta^{(t)}$ with $\alpha$ SGD steps to get $\theta^{(t+1)}$
7:     Update $w^{(t+1)}$ from $w^{(t)}$ according to equations (5) and (6)
8: **end for**
9: **return** $(Q_1, w_1^{(S)}), (Q_2, w_2^{(S)}), \ldots, (Q_m, w_m^{(S)})$; $\theta^{(S)}$

---

In Appendix 2, we provide a generalization bound to justify the consistency of this procedure. We demonstrate that the generalization error of the model trained by Algorithm 2 is bounded by a bias increase term from transferring from across different groups, and a variance reduction term from data augmentation. The proof technique involves carefully analyzing the bilevel optimization algorithm using covering numbers, which builds on the transfer exponent framework of Chen et al. (2022) and the work of Hanneke & Kpotufe (2019).

### 3.2.2 Running Time Analysis

Next, we examine the runtime of the weighted training step. Compared to SGD, the algorithm includes updating the group weights using equations (5) and (6). This step involves computing Hessian inverse and the product between Hessian inverse with the gradient of each group. To avoid explicitly computing Hessian inverse, we use the following method:

- First, estimate the inverse Hessian-gradient vector $s := H^{-1}v$, where $v = \sum_{g=1}^{m} q_g \nabla_\theta \hat{L}_g(f_{\theta^w})$. We apply conjugate gradient with stochastic estimation. We samples $n$ data points $\{(x_j, y_j)\}_{j=1}^{n}$, recursively computing

$$A_j v = v + (I - \nabla_\theta^2 L(f_\theta(x_j), y_j)) A_{j-1} v,$$

and takes $A_n v$ as the final estimate of $H^{-1}v$. For $n$ data points and $\theta \in \mathbb{R}^p$, this procedure takes $O(np)$ time.

- Secondly, compute $d_i = -s^\top \nabla_\theta \hat{L}_i(f_{\theta^w}; Q_i)$. Since there are $m$ groups, this step takes $O(mp)$ time.

Taken together, updating the group weights takes $O(np + mp)$ time. In our implementation, we conduct the above estimation using one batch of $b$ data points from each group, leading to a running time of $O(bmp + mp) = O(bmp)$.

Empirically, we find that the running time is comparable to SGD. To illustrate, we train a three-layer graph neural network for protein graph classification. Running the above update takes 1.8 GPU hours, which is $1.1\times$ of SGD, which takes 1.6 GPU hours. On another image classification data set for training ResNet-50, the runtime is 0.8 GPU hours, $1.3\times$ of SGD, which takes 0.6 hours.

Lastly, we illustrate the convergence behavior in Figure 6. The loss curves on both data sets behave similarly to SGD, with weighted training achieving a lower test loss.

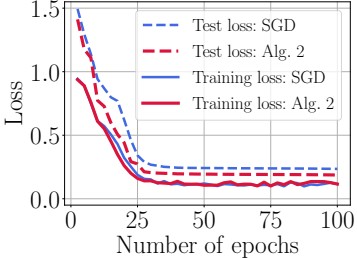 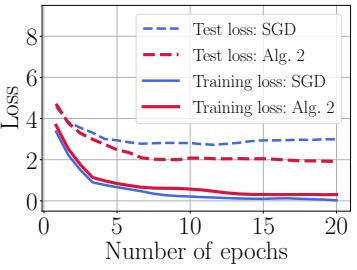

(a) Training a three-layer GNN for graph classification  (b) Training a ResNet-50 for image classification

Figure 6: Showing the loss curves of both weighted training and SGD. The left figure shows the results of training a three-layer neural network on protein graph classification. The right figures show the result of training a ResNet-50 on image classification.

## 4  Experiments

In this section, we evaluate the performance of Algorithm 2 for numerous data sets in terms of both the runtime and the performance of the augmentation. We will consider both supervised and self-supervised learning settings. In particular, we will compare our method with existing optimization methods for augmentation training. We will also compare with schemes designed based on domain knowledge, such as RandAugment (Cubuk et al., 2020) and SimCLR (Chen et al., 2020b). The results are broadly relevant to learning with little labeled data, with a particular focus on graphs and images. We summarize a few key findings as follows:

- In supervised learning settings with both graph data sets (such as protein graphs) and image data sets (such as medical images), our approach can outperform augmentation search methods on the graph classification data set by **1.9**% with **43**% less runtime and on the image classification data set by **3.0**% with **38**% less runtime. Compared to RandAugment which does not search for a composition, we note **4.3**% and **5.7**% better performance, respectively.

- For contrastive learning settings, we can now get up to **7.4%** improvement across eight graph and image data sets. The runtime is also reduced by **32**%. Compared to SimCLR (see Figure 3), the augmentation scheme can deliver comparable performance on CIFAR-10. On a medical image classification task, we now find a new scheme that outperforms SimCLR by **5.9**%.

- We verify that the improvement is statistically significant, by conducting the Wilcoxon signed-rank test on the performance of our algorithm compared to the baselines across ten datasets. When comparing our algorithm with the recent augmentation search method and with RandAugment, the test yields $p$-values of 0.0025 and 0.0019, respectively. This indicates that the probability of our algorithm showing no improvement is less than 1%.

- Lastly, we provide ablation studies to justify the design, including finding the tree for each group and the weighted training, for achieving the final performance.

Our code for reproducing the experiments is available at https://github.com/VirtuosoResearch/Tree-data-augmentation, which also includes instructions for loading the new dataset.

## 4.1 Experimental Setup

We apply our algorithm to two settings. The first setting is supervised learning, which uses data augmentation to generate labeled examples. The second setting is contrastive learning, which uses data augmentation to generate contrastive examples.

**Data sets.** For supervised learning, we construct a graph classification data set for protein function prediction collected from the AlphaFold Protein Structure Database.[1] This data set includes 20,504 human proteins structured as undirected graphs, where nodes correspond to amino acid residues labeled with one of 20 amino acid types. The edges are the spatial distances between two amino acids thresholded at 6Å between two $C\alpha$ atoms. The problem is multi-label classification containing 1,198 protein functions, each function viewed as a binary label. We split the data set into 16 groups using intervals of graph sizes and average degrees. The idea is to group graphs of similar sizes together. To determine the optimal number of groups, we experiment with dividing the graphs into $4, 9, 16,$ and $25$ groups, corresponding to splitting graph sizes and average degrees into $2, 3, 4,$ and $5$ intervals, respectively. Increasing $m$ beyond 16 did not lead to any benefit. Therefore, we restrict $m$ to less than 16.

The main difference between our data set and previous data sets (Borgwardt et al., 2005; Hu et al., 2020a) is the coverage of protein functions. The data set from Borgwardt et al. (2005) predicts whether a protein is an enzyme as a binary classification task. The data set from Hu et al. (2020a) classifies a protein into 37 taxonomic groups as different species as a multiclass classification problem. Our data set involves 1,198 protein functions using Gene Ontology (GO) annotation. Another difference is in constructing the graphs. Borgwardt et al. (2005) and our data set abstract the structure of the protein itself as a graph. Hu et al. (2020a) constructs the graph based on a relationship between proteins in protein association networks. The third difference is that the graphs in our data set exhibit more significant variations in their sizes, ranging from 16 to 34, 350. This comparison is summarized in Table 3.

Table 3: Comparison of our protein graph classification data set to two existing data sets. Our data set is constructed from a newer data source and covers a broader range of (protein) functions than previous ones.

|  | PROTEINS | OGBG-PPA | Our Data Set |
|---|---|---|---|
| # Graphs | 1113 | $158, 100$ | $20, 504$ |
| # Functions | 2 | 37 | $1, 198$ |
| # Nodes | $4 \sim 620$ | $50 \sim 300$ | $16 \sim 34, 350$ |
| Avg. Degree | $3.4 \sim 10.1$ | $2.0 \sim 240.9$ | $1.8 \sim 9.4$ |
| Category | Binary classification: whether or not the protein is an enzyme | Multi-class classification for 37 species, e.g., mammals, bacterial families, archaeans | Multi-label classification for 1,198 GO terms that describe biological functions of proteins |
| Graph Type | Attributed and undirected graphs, where nodes represent secondary structure elements (SSE), and edges represent spatial closeness | Undirected graphs of protein associations, where nodes represent proteins and edges indicate biologically meaningful associations between proteins. | Undirected graphs, where nodes represent the amino acids and edges represent their spatial distance thresholded at 6Å between two $C\alpha$ atoms |
| Data Source | Protein data bank | Protein association network | AlphaFold APIs |

Next, we consider an image classification task using the iWildCam data set from the WILDS benchmark (Beery et al., 2021), where images are collected from different camera traps with varied illumination, camera angles, and backgrounds. The problem is multi-class classification: given a camera trap image, predict the labels as one of 182 animal species. We evaluate the macro-$F_1$ score on this data set, which is the averaged $F_1$ score over every class. We consider three groups from three cameras with the most images.

---

[1]https://alphafold.ebi.ac.uk/

For contrastive learning, we consider image classification, including CIFAR-10 and a medical image data set containing eye fundus images for diabetic retinopathy classifications. This is a multi-class classification problem: given an eye fundus image, predict the severity of diabetic retinopathy into five levels. We consider three groups based on three hospitals, which differ in the patient population and imaging devices. The sources are available online: Messidor,[2] APTOS,[3] and Jinchi[4].

We also consider six graph classification data sets from TUdatasets (Morris et al., 2020), including NCI1, Proteins, DD, COLLAB, REDDIT, and IMDB, for testing graph contrastive learning. The first three data sets contain small molecular graphs where nodes represent atoms and edges represent chemical bonds. The goal is to predict the biological properties of molecular graphs. The latter three data sets contain social networks abstracted from scientific collaborations, online blogs, and actor collaborations. The goal is to predict the topic of the network. We split each data set into four groups by their sizes and average degrees.

**Baselines.** We compare our approach with pre-specified data augmentation methods, including ones designed for specific group shifts, and optimization methods that search for compositions. For supervised learning, we consider RandAugment (Cubuk et al., 2020), Learning Invariant Predictors with Selective Augmentation (LISA) (Yao et al., 2022), and targeted augmentations (Gao et al., 2023). We also consider optimization methods including GraphAug (Luo et al., 2023) on graph data sets and Stochastic Compositional Augmentation Learning (SCALE) (Chatzipantazis et al., 2023) on image data sets.

For contrastive learning, we consider SimCLR (Chen et al., 2020b) and LISA (Yao et al., 2022) on image data sets, InfoGraph (Sun et al., 2020), RandAugment (Cubuk et al., 2020), and GraphCL (You et al., 2020) on graph data sets. We also consider Joint Augmentation Optimization (JOAO) (You et al., 2021) on graph data sets, and SCALE (Chatzipantazis et al., 2023) on image data sets.

**Implementations.** We use four graph transformation functions from You et al. (2020), including DropNodes, PermuteEdges, Subgraph, and MaskingNodes. Each method modifies a fraction of nodes, edges, or node features. Specifically, DropNodes deletes a randomly sampled set of nodes and their connections. PermuteEdges adds and deletes a randomly sampled set of edges. Subgraph extracts a subgraph generated by random walks from randomly sampled nodes in the graph. NodeMasking masks a randomly sampled fraction of node attributes to zero. We use $0.1, 0.2, 0.3, 0.4, 0.5$ as their perturbation magnitudes. In total, $k = 20$.

For images, we use sixteen image transformations including ShearX/Y, TranslateX/Y, Rotate, AutoContrast, Invert, Equalize, Solarize, Posterize, Contrast, Color, Brightness, Sharpness, Cutout, and Sample Pairing. We consider five values of perturbation magnitude uniformly spaced between intervals. In total, $k = 80$.

We use a three-layer graph neural network on graph data sets. We use pretrained ResNet-50 on image data sets. In terms of hyperparameters, we search the maximum depth $d$ up to 4 and $H$ between $[0, 1]$. For weighted training, we adjust the learning rate $\eta$ between $0.01, 0.1, 1.0$ and the SGD steps $\alpha$ between $25, 50, 100$.

## 4.2 Experimental Results

**Supervised learning.** We first discuss the results in supervised learning settings, as reported in Table 4. We compare our algorithm with two types of baselines, including pre-specified augmentation methods and optimization methods that search for the composition.

On the protein graph classification data set, our algorithm outperforms RandAugment, which randomly samples a composition of augmentations for each batch of data, and LISA, which applies the mixup technique between input examples, by **4.3**% and **2.3%**, respectively. This shows the benefit of searching augmentations over pre-specified schemes. In addition, we also see an improvement of **1.9**% over GraphAug.

---

[2] https://www.adcis.net
[3] https://kaggle.com/competitions/aptos2019
[4] https://figshare.com

Table 4: We compare our algorithm with several existing data augmentation schemes on a protein graph classification data set (left) and a wildlife image classification data set (right). In particular, the left-hand side shows the average test AUROC scores for protein function prediction. The right shows the test macro $F_1$ score on the image classification data set. We report the averaged results over five random seeds.

|  | Graph Classification | Image Classification |
| --- | --- | --- |
| Training Set Size | 12,302 | 6,568 |
| Validation Set Size | 4,100 | 426 |
| Testing Set Size | 4,102 | 789 |
| # Classes | 1,198 | 182 |
| Metric | AUROC | Macro $F_1$ |
| Empirical Risk Min. | $71.3 \pm 0.3$ | $52.4 \pm 1.1$ |
| RandAugment | $71.8 \pm 0.8$ | $58.9 \pm 0.4$ |
| LISA | $73.2 \pm 0.3$ | $59.7 \pm 0.3$ |
| Targeted Augmentation | - | $56.3 \pm 0.4$ |
| GraphAug | $73.5 \pm 0.3$ | - |
| SCALE | - | $60.4 \pm 0.5$ |
| **Algorithm 2 (Ours)** | $\mathbf{74.9 \pm 0.4}$ | $\mathbf{62.3 \pm 0.6}$ |

On the image classification data set, our algorithm outperforms pre-specified data augmentation methods, including RandAugment, targeted augmentation, and LISA, by **6.8%** on average. The improvement over SCALE is about **3.0**%.

Next, we report the runtime. Recall that we first find augmentation composition in each group and then learn a weighted combination of them. This algorithm takes 8.2 hours on the graph data set and 2.6 hours on the image data set. This is **44**% and **38**% less than the optimization methods on the graph and image data sets, which use 14.4 and 4.2 hours, respectively. Moreover, as discussed in Section 3.2, the weighted training procedure of our algorithm takes a comparable runtime as SGD.

**Contrastive learning.** Next, we report the results of contrastive learning. Again, we consider pre-specified augmentation methods, such as SimCLR, and existing optimization methods. We evaluate trained models by linear evaluation using an SVM classifier on the contrastive features.

We first report the results of contrastive learning on image data sets in Table 5. On CIFAR-10, which contains natural images, our algorithm performs on par with SimCLR designed by domain experts. On a medical image data set, we observe that the SimCLR augmentation scheme does not work on medical images and even decreases the performance compared to just using the pretrained network's features. By finding compositions of augmentations, our algorithm improves over SimCLR by **5.9**%. In both data sets, our algorithm outperforms RandAugment by **2.4**% on average.

Moreover, our algorithm improves an optimization method, SCALE, by **1.1**% on both image data sets on average. Our algorithm also takes **32**% less runtime than the optimization method.

For graph contrastive learning, we report the 10-fold cross-validation results in Table 6. Compared to pre-specified schemes, our algorithm outperforms RandAugment by up to **20**% and GraphCL by up to **17**% across six data sets. Compared to JOAO, our algorithm improves by **7.1**%. Our algorithm benefits from splitting graphs into groups with similar sizes and degrees before applying graph contrastive learning.

**Ablation studies.** We verify that both parts of the algorithm contribute to the results, including finding one tree per group and weighted training. First, we remove the tree for each group and instead find a single tree for all groups. This resulted in a performance decrease of 0.6% and 1.0% on the protein graph and image classification data set, respectively. Second, we remove weighted training and replace it with uniform weighting. This leads to a performance decrease of 1.0% and 1.6%, respectively. From analyzing model features, we find that our approach results in more similar features between groups than a single tree. For details, see Appendix C.

Table 5: We compare our algorithm with several augmentation methods for contrastive learning on images. We report the test accuracy on CIFAR-10 and a medical image classification data set (Messidor). We conduct contrastive learning using ResNet-50 pretrained on ImageNet and evaluate the test accuracy using linear evaluations of last-layer features. The results are averaged over five random seeds.

|  | Natural Image Classification | Medical Image Classification |
|---|---|---|
| Training Set Size | 45,000 | 10,659 |
| Validation Set Size | 5,000 | 2,670 |
| Test Set Size | 5,000 | 3,072 |
| # Classes | 10 | 5 |
| Pretrained Features | 88.9% $\pm$ 0.0 | 70.1% $\pm$ 0.0 |
| SimCLR | **92.8**% $\pm$ 0.5 | 69.1% $\pm$ 1.1 |
| RandAugment | 89.8% $\pm$ 0.7 | 72.4% $\pm$ 0.5 |
| LISA | - | 72.1% $\pm$ 0.5 |
| SCALE | 91.3% $\pm$ 0.3 | 72.4% $\pm$ 0.2 |
| **Algorithm 2 (Ours)** | **93.0**% $\pm$ 0.4 | **73.2**% $\pm$ 0.3 |

Table 6: We compare our algorithm with several existing augmentation methods in graph contrastive learning. We report the test classification accuracy on several graph prediction data sets. We first conduct contrastive learning using three-layer graph neural networks and evaluate the test accuracy using linear evaluations of last-layer features. The results are based on ten-fold cross-validation.

|  | NCI1 | PROTEINS | DD | COLLAB | REDDIT-B | IMDB-B |
|---|---|---|---|---|---|---|
| # Graphs | 4,110 | 1,113 | 1,178 | 5,000 | 2,000 | 1,000 |
| # Classes | 2 | 2 | 2 | 3 | 2 | 2 |
| InfoGraph | 76.2% | 74.4% | 72.8% | 70.6% | 82.5% | 73.0% |
| RandAugment | 62.0% | 72.2% | 75.7% | 58.1% | 76.3% | 55.2% |
| GraphCL | 77.8% | 74.3% | 78.6% | 71.3% | **89.4**% | 71.1% |
| JOAO | 78.0% | 74.5% | 77.4% | 69.5% | 86.4% | 70.8% |
| **Algorithm 2 (Ours)** | **79.3**% | **77.7**% | **78.9**% | **78.4**% | **89.4**% | **74.4**% |

In Algorithm 1, we require the maximum depth of the tree $d$ and the probabilities $H$. In Algorithm 2, we require the number of SGD steps $\alpha$, and learning rate $\eta$ for updating combination weights. In each ablation study, we vary one hyper-parameter while keeping the others constant. The fixed hyperparameters are as follows: tree depth of 4, number of probabilities of 10, SGD steps of 50, and learning rate of 0.1. Table 7 presents the findings. Setting $d = 4$ and $|H| = 10$ yields the best results. Additionally, using $\alpha = 50$ and $\eta = 0.1$ yields the best results. We also note that these are effective in other settings. Therefore, we use these values as the default in all of our experiments.

## 4.3 Extension and Discussion

**Extension.** To further illustrate the application of our approach, we consider semi-supervised learning and use tree augmentation for consistency regularization (Xie et al., 2020).

We evaluate our approach on CIFAR-10 and SVHN. For CIFAR-10, we use 1,000 labeled and 44,000 unlabeled examples. For SVHN, we use 1,000 labeled and 64,932 unlabeled examples. we also consider Chest-X-Ray, which contains frontal-view X-ray images labeled with 14 diseases, each viewed as a zero-one label (Wang et al., 2017; Rajpurkar et al., 2017). We use 1,000 labeled and 9,000 unlabeled examples. We train a randomly initialized Wide-ResNet-28-10 on all datasets using SGD with a learning rate of 0.03 and 100,000 gradient update steps, following Xie et al. (2020). To determine the composition of data augmentation, we use the same 16 types of transformations as in other image data sets.

Table 7: Ablation study of varying maximum tree depth $d$, number of probability values $|H|$, SGD steps $\alpha$, and learning rate $\eta$ in Algorithm 2. We report the average AUROC for the protein graph classification data set and the Macro-$F_1$ for the wildlife image classification data set (on validation sets). The results are averaged over five random seeds.

| Graph Classification (AUROC) | | | Image Classification (macro $F_1$) | | |
|---|---|---|---|---|---|
| $d = 2$ | $d = 3$ | $\boldsymbol{d = 4}$ | $d = 2$ | $d = 3$ | $\boldsymbol{d = 4}$ |
| $72.4\% \pm 0.1$ | $74.2\% \pm 0.2$ | $\mathbf{74.5\%} \pm 0.4$ | $61.6\% \pm 0.6$ | $65.0\% \pm 0.3$ | $\mathbf{65.3\%} \pm 0.6$ |
| $|H| = 5$ | $\boldsymbol{|H| = 10}$ | $|H| = 20$ | $|H| = 5$ | $\boldsymbol{|H| = 10}$ | $|H| = 20$ |
| $72.9\% \pm 0.2$ | $\mathbf{74.5\%} \pm 0.4$ | $\mathbf{74.5\%} \pm 0.6$ | $64.7\% \pm 0.3$ | $\mathbf{65.3\%} \pm 0.6$ | $\mathbf{65.3\%} \pm 0.4$ |
| $\alpha = 25$ | $\boldsymbol{\alpha = 50}$ | $\alpha = 100$ | $\alpha = 25$ | $\boldsymbol{\alpha = 50}$ | $\alpha = 100$ |
| $74.0\% \pm 0.2$ | $\mathbf{74.5\%} \pm 0.4$ | $73.4\% \pm 0.8$ | $64.8\% \pm 0.3$ | $\mathbf{65.3\%} \pm 0.6$ | $64.2\% \pm 0.4$ |
| $\eta = 1.0$ | $\boldsymbol{\eta = 0.1}$ | $\eta = 0.01$ | $\eta = 1.0$ | $\boldsymbol{\eta = 0.1}$ | $\eta = 0.01$ |
| $71.0\% \pm 0.1$ | $\mathbf{74.5\%} \pm 0.4$ | $72.4\% \pm 0.8$ | $60.5\% \pm 0.7$ | $\mathbf{65.3\%} \pm 0.6$ | $64.3\% \pm 0.5$ |

Table 8 reports the test performance on three data sets. We verify that our algorithm can outperform UDA, which uses RandAugment, by **1.2**% averaged over the data sets.

Table 8: We apply our approach for semi-supervised learning. We compare to Unsupervised Data Augmentation (UDA) (Xie et al., 2020), which uses RandAugment as the base augmentation scheme. We report the results from training a randomly initialized WideResNet-28-10, averaged over five random seeds.

| | CIFAR-10 | SVHN | Chest-X-Ray |
|---|---|---|---|
| Labeled Training Set Size | 1,000 | 1,000 | 1,000 |
| Unlabeled Training Set Size | 44,000 | 64.932 | 9,000 |
| Validation Set Size | 5,000 | 7,325 | 7,846 |
| Test Set Size | 10,000 | 26,032 | 22,433 |
| Metrics | Error Rates ($\downarrow$) | Error Rates ($\downarrow$) | AUROC ($\uparrow$) |
| UDA | $4.9\% \pm 0.1$ | $2.7\% \pm 0.1$ | $73.3\% \pm 0.5$ |
| **Algorithm 1 (Ours)** | $\mathbf{4.5\%} \pm 0.1$ | $\mathbf{2.2\%} \pm 0.1$ | $\mathbf{75.2\%} \pm 0.6$ |

**Discussion.** One justification for our empirical findings is that the trees generalize the sequences, which have been the focus of prior work (Ratner et al., 2017; Wu et al., 2020; Xie et al., 2020; Chen et al., 2020b). Since the search now happens in this generalized set, the results are expected to be on par, if not better. It is an interesting research question to understand why the greedy construction of trees works well empirically. See a recent work (Deng & Hsu, 2024) that develops a new framework for formally reasoning about learning hierarchical groups in multi-group learning.

## 5  Conclusion

This paper designs an algorithm for learning a binary tree composition of data augmentation. The algorithm uses a top-down recursive search method to find a tree with reduced running time complexity. Experiments validate that the algorithm reduces the runtime compared to existing search methods without decreasing downstream performance. Next, the algorithm is extended to the case of learning under group shifts. The algorithm first finds one tree for each group to account for heterogeneous features and then reweights each tree into a forest of trees. Extensive experiments show the empirical benefits of this approach over existing augmentation methods. The algorithm can be readily extended to tackle different loss metrics such as the worst-group loss. It may also be worth revisiting out-of-domain generalization where heterogeneous domains are present through the design of automatic data augmentation.

## Acknowledgement

We would like to thank Michelle Velyunskiy for her work on collecting the protein graphs data set during the initial stage of this project. Thanks to Jinhong Yu for setting up some of the computational environments in the experiments. We are very grateful to the anonymous reviewers and the action editor for various insightful comments that have resulted in a significant improvement of this work.

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

## A    Derivation of Equation (5)

By regarding $\theta^w$ as a function of $w$ and applying the chain rule, we have the following:

$$\frac{\partial \hat{L}(f_{\theta^w})}{\partial w} = \left(\nabla_\theta \hat{L}(f_{\theta^w})\right)^\top \frac{\partial \theta^w}{\partial w}. \tag{7}$$

By the definition of $\theta^w$, we must have that $\theta^w$ is a minimizer of $\sum_{g=1}^m w_g \cdot \hat{L}_g(f_\theta; Q_g)$. Thus, the gradient with respect to $\theta$ must be zero, given $w$. Next, we write the gradient as:

$$F(w, \theta) = \nabla_\theta \sum_{g=1}^m w_g \cdot \hat{L}_g(f_\theta; Q_g) = \sum_{g=1}^m w_g \cdot \nabla_\theta \hat{L}_g(f_\theta; Q_g). \tag{8}$$

For a stationary point $(w, \theta^w)$ of equation (4), we have $F(w, \theta^w) = 0$. By the implicit function theorem,[5] if $F(w, \theta)$ is differentiable and $\frac{\partial F(w,\theta)}{\partial \theta}$ is invertible, then for a $(w, \theta)$ near some $(\tilde{w}, \tilde{\theta})$ satisfying $F(\tilde{w}, \tilde{\theta}) = 0$, there exists a function mapping $w \to \theta^w$ which is continuous and continuously differentiable and the derivative of this mapping over $w$ is given by:

$$\frac{\partial \theta^w}{\partial w} = -\left(\left.\frac{\partial F(w, \theta)}{\partial \theta}\right|_{\theta^w}\right)^{-1} \left(\left.\frac{\partial F(w, \theta)}{\partial w}\right|_{\theta^w}\right). \tag{9}$$

Replacing equation (9) into equation (7), we get

$$\frac{\partial \hat{L}(f_{\theta^w})}{\partial w_i} = -\left(\sum_{g=1}^m q_g \nabla_\theta \hat{L}_g(f_{\theta^w}; Q_g)\right)^\top \left(\sum_{g=1}^m w_g \nabla_\theta^2 \hat{L}_g(f_{\theta^w}; Q_g)\right)^{-1} \nabla_\theta \hat{L}_i(f_{\theta^w}; Q_i), \tag{10}$$

which concludes the proof of equation (5).

## B    Consistency of the Weighted Training Procedure

This section derives a generalization bound for the bilevel optimization algorithm. The result shows that when the number of samples goes to infinity, the gap between training and test errors will shrink to zero, plus a discrepancy distance term between the source and target tasks. Let $\{P_g\}_{g=1}^m$ denote the $m$ groups of augmented data distributions. Concretely, for group $g$, we have $n_g$ i.i.d samples. Let $\hat{P}_g = (x_{g,i}, y_{g,i})_{i=1}^{n_g}$ denote the augmented set, by applying $Q_g$ to the input samples from $P_g$. We denote $P_0$ as a target data distribution. Given a bounded loss function $\ell$, let $L_g(f_\theta)$ and $\hat{L}_g(f_\theta)$ denote the expected and empirical loss on group $g$, respectively. We denote the minimizer of the weighted empirical loss as

$$\hat{\theta} \in \arg\min_{\theta \in \Theta} \sum_{g=1}^m w_g \hat{L}_g(f_\theta). \tag{11}$$

We denote the optimal representation as:

$$\theta_0^\star \in \arg\min_{\theta_0 \in \Theta} L_0(\theta_0).$$

We denote the $w$-weighted optimal representation as:

$$\bar{\theta}^w \in \arg\min_{\theta \in \Theta} \sum_{g=1}^m w_g L_g(\theta).$$

Since $\bar{\theta}^w$ may not be unique, we introduce the function space $\bar{\theta}^w \in \bar{\Theta}^w$ to include all $w$-weighted $\bar{\theta}^w$.

---

[5] https://en.wikipedia.org/wiki/Implicit_function_theorem

**Definition B.1** $((\rho, C_\rho)$-transferable$)$**.** *A representation $\theta \in \Theta$ is $(\rho, C_\rho)$-transferable from $w$-weighted source subsets to the target subset, if there exists $\rho > 0, C_\rho > 0$ such that for any $\bar{\theta}^w \in \bar{\Theta}^w$, we have*

$$L_0(\theta) - L_0(\bar{\theta}^w) \leq C_\rho \Big( \sum_{g=1}^{m} w_g \left( L_g(\theta) - L_g(\bar{\theta}^w) \right) \Big)^{\frac{1}{\rho}}.$$

Next, we state two technical assumptions. The first assumption describes the loss function's Lipschitz continuity.

**Assumption B.1.** *Let the loss function $\ell : \mathcal{X} \times \mathcal{Y} \to [0, 1]$ be $C$-Lipschitz in $x \in \mathcal{X}$, meaning for all $x_1, x_2 \in \mathcal{X}$ and $y \in \mathcal{Y}$, the following holds*

$$|\ell(x_1, y) - \ell(x_2, y)| \leq C \cdot \|x_1 - x_2\|.$$

The second assumption describes the covering size of the functional class.

**Assumption B.2.** *There exist constants $C_\Theta$ and $v_\Theta$ greater than 0 such that for any probability measure $\mathbb{Q}_\mathcal{X}$ on $\mathcal{X} \subseteq \mathbb{R}^d$, we have*

$$\mathcal{N}\Big(\epsilon; \Phi; L^2(\mathbb{Q}_\mathcal{X})\Big) \leq \left( \frac{C_\Theta}{\epsilon} \right)^{v_\Theta} \quad \text{for any } \epsilon > 0,$$

*where $\mathcal{N}\big(\epsilon; \Phi; L^2(\mathbb{Q}_\mathcal{X})\big)$ is the minimum number of $L^2(\mathbb{Q}_\mathcal{X})$ balls with radius $\epsilon$ required to cover the entire space. Here, the $L^2(\mathbb{Q}_\mathcal{X})$ distance between two vector-valued functions $\theta$ and $\psi$ is defined as*

$$L^2(\mathbb{Q}_\mathcal{X}) = \left( \int \|\theta(x) - \psi(x)\|^2 \, d\mathbb{Q}_\mathcal{X}(x) \right)^{\frac{1}{2}}.$$

Given a weight vector $w$, let us define

$$\text{dist}\Big( \sum_{g=1}^{m} w_g P_g, P_0 \Big) := \sup_{\bar{\theta}^w \in \bar{\Theta}^w} L_0(\bar{\theta}^w) - L_0(\theta_0^\star),$$

which represents the distance between the source $w$-weighted and target groups. Now, we are ready to state the result.

**Theorem B.3.** *Let $\hat{\theta}, \bar{\theta}^w, \theta_0^\star$ be defined as the above equations with a fixed $w$. Suppose Assumptions B.1 and B.2 both hold. Let $\delta \in (0, 1)$ be a fixed real number. Then, for any representation $\hat{\theta} \in \bar{\Theta}^w$ such that $\hat{\theta}$ is $(\rho, C_\rho)$-transferable, with probability at least $1 - \delta$, we have:*

$$L_0(\hat{\theta}) - L_0(\theta_0^\star) \leq \text{dist}\left( \sum_{g=1}^{m} w_g P_g, P_0 \right) + C_\rho \left( \frac{C_1 \sqrt{v_\Theta \log(C_\Theta C)} + C_2 \sqrt{\log(\delta^{-1})}}{\sqrt{N_w}} \right)^{\frac{1}{\rho}},$$

*where $N_w = \big( \sum_{g=1}^{m} \frac{w_g^2}{n_g} \big)^{-1}$, $C_1$ and $C_2$ are two constants that do not grow with the size of the input.*

At a high level, the first distance term represents the bias increase by transferring across groups, and the second variance term is the empirical error of learning an imperfect representation $\bar{\theta}^w$. This term reduces under pooling; Consider a scenario where $n_g$ is the same across groups, then through $N_w$, the second term reduces by a factor of $m^{-1}$. The proof technique is based on carefully examining the weighted training procedure using covering numbers (Chen et al., 2022; Hanneke & Kpotufe, 2019).

*Proof.* Let $\Theta$ be the domain for which $\theta$ lies in. Consider a bounded loss function $\ell$. To establish the result, we begin by decomposing the difference between $L_0(\hat{\theta})$ and $L_0(\bar{\theta}^w)$. We proceed with the following calculations:

$$L_0(\hat{\theta}) - L_0(\bar{\theta}^w) = \underbrace{L_0(\hat{\theta}) - L_0(\bar{\theta}^w) + L_0(\bar{\theta}^w) - L_0(\theta_0^\star)}_{\text{adding and subtracting } L_0(\bar{\theta}^w)}$$

$$\leq C_\rho \left( \sum_{g=1}^m \omega_g (L_g(\hat{\theta}) - L_g(\bar{\theta}^w)) \right)^{\frac{1}{\rho}} + \underbrace{\sup_{\bar{\theta}^w \in \bar{\Theta}^w} \left( L_0(\bar{\theta}^w) - L_0(\theta_0^\star) \right)}_{\text{upper bounded by the supremum}} \tag{12}$$

$$= C_\rho \left( \sum_{g=1}^m \omega_g (L_g(\hat{\theta}) - L_g(\bar{\theta}^w)) \right)^{\frac{1}{\rho}} + \text{dist} \left( \sum_{g=1}^m \omega_g P_g, P_0 \right).$$

Now we focus on the first term from above, which is $\sum_{g=1}^m \omega_g (L_g(\hat{\theta}) - L_g(\bar{\theta}^w))$. We have

$$\sum_{g=1}^m \omega_g (L_g(\hat{\theta}) - L_g(\bar{\theta}^w)) = \sum_{g=1}^m \omega_g \left( L_g(\hat{\theta}) - \hat{L}_g(\hat{\theta}) + \hat{L}_g(\hat{\theta}) - \hat{L}_g(\bar{\theta}^w) + \hat{L}_g(\bar{\theta}^w) - L_g(\bar{\theta}^w) \right)$$

$$\leq \sum_{g=1}^m \omega_g \left( (L_g(\hat{\theta}) - \hat{L}_g(\hat{\theta}) + \hat{L}_g(\bar{\theta}^w) - L_g(\bar{\theta}^w) \right) \tag{13}$$

$$\leq \sup_{\theta \in \Theta} \left( \sum_{g=1}^m \omega_g \left( (L_g(\theta) - \hat{L}_g(\theta) + \hat{L}_g(\bar{\theta}^w) - L_g(\bar{\theta}^w)) \right) \right)$$

$$= \sup_{\theta \in \Theta} \left( \sum_{g=1}^m \omega_g \cdot \frac{1}{n_g} \sum_{i=1}^{n_g} \left( L_g(\theta) - \ell(f_\theta(\mathbf{x}_{g,i}), y_{g,i}) + \ell(f_{\bar{\theta}^w}(\mathbf{x}_{g,i}), y_{g,i}) - L_g(\bar{\theta}^w) \right) \right).$$

Denote the last equation above as $G(\{\mathbf{z}_{g,i}\})$, where $\mathbf{z}_{g,i} = \{\mathbf{x}_{g,i}, y_{g,i}\}$. Step (13) is by noting that $\hat{L}_g(\hat{\theta}) - \hat{L}_g(\bar{\theta}^w)$ is at most zero according to equation (11).

Next, we will apply McDiarmid's inequality. Let us fix two indices $1 \leq g' \leq m$ and $1 \leq i_{g'} \leq n_g$. Let us define $\{\tilde{\mathbf{z}}_{g',i}\}$ by replacing $\mathbf{z}_{g',i_{g'}}$ with another $\tilde{\mathbf{z}}_{g',i_{g'}} = (\tilde{\mathbf{x}}_{g',i_{g'}}, \tilde{y}_{g',i_{g'}}) \in \mathcal{X} \times \mathcal{Y}$. Given that $\{\mathbf{z}_{g,i}\}$ and $\{\tilde{\mathbf{z}}_{g',i}\}$ differ by only one element, we have

$$|G(\{\mathbf{z}_{g,i}\}) - G(\{\tilde{\mathbf{z}}_{g',i}\})| = \left| \sum_{g \neq g'} \sum_{i=1}^{n_g} \frac{\omega_g}{n_g} \left( L_g(\theta) - \ell(f_\theta(\mathbf{x}_{g,i}), y_{g,i}) + \ell(f_{\bar{\theta}^w}(\mathbf{x}_{g,i}), y_{g,i}) - L_g(\bar{\theta}^w) \right) \right.$$

$$+ \sum_{i \neq i_{g'}} \frac{\omega_{g'}}{n_{g'}} \left( L_{g'}(\theta) - \ell(f_\theta(\mathbf{x}_{g',i}), y_{g',i}) + \ell(f_{\bar{\theta}^w}(\mathbf{x}_{g',i}), y_{g',i}) - L_{g'}(\bar{\theta}^w) \right)$$

$$\left. + \frac{\omega_{g'}}{n_{g'}} \left( L_{g'}(\theta) - \ell(f_\theta(\mathbf{x}_{g',i_{g'}}), y_{g',i_{g'}}) + \ell(f_{\bar{\theta}^w}(\mathbf{x}_{g',i_{g'}}), y_{g',i_{g'}}) - L_{g'}(\bar{\theta}^w) \right) \right|$$

$$\leq \frac{\omega_{g'}}{n_{g'}} \left( \left| L_{g'}(\theta) - \ell(f_\theta(\mathbf{x}_{g',i_{g'}}), y_{g',i_{g'}}) \right| + \left| L_{g'}(\bar{\theta}^w) - \ell(f_{\bar{\theta}^w}(\mathbf{x}_{g',i_{g'}}), y_{g',i_{g'}}) \right| \right.$$

$$\left. + \left| L_{g'}(\theta) - \ell(f_\theta(\tilde{\mathbf{x}}_{g',i_{g'}}), \tilde{y}_{g',i_{g'}}) \right| + \left| L_{g'}(\bar{\theta}^w) - \ell(f_{\bar{\theta}^w}(\tilde{\mathbf{x}}_{g',i_{g'}}), \tilde{y}_{g',i_{g'}}) \right| \right) \leq \frac{4\omega_{g'}}{n_{g'}}.$$

The last step above is based on the fact that the loss function is bounded between 0 and 1. Thus, we derive the following result:

$$\Pr \left( G(\{\mathbf{z}_{g,i}\}) - \mathbb{E}[G(\{\mathbf{z}_{g,i}\})] \geq \epsilon \right) \leq \exp \left( -\frac{2\epsilon^2}{\sum_{g=1}^m \sum_{i=1}^{n_g} \frac{16\omega_g^2}{n_g^2}} \right), \; \forall \epsilon > 0. \tag{14}$$

Recall our previous definition that $N_w = \left( \sum_{g=1}^m \frac{\omega_g^2}{n_g} \right)^{-1}$. By setting $\delta$ as

$$\delta = \exp\left( -\frac{2\epsilon^2}{\sum_{g=1}^m \sum_{i=1}^{n_g} \frac{16\omega_g^2}{n_g^2}} \right),$$

we get that $\epsilon = 2\sqrt{2}\sqrt{\frac{\log(1/\delta)}{N_w}}$. Thus, Equation (14) can be equivalently stated as:

$$G(\{\mathbf{z}_{g,i}\}) \le \mathbb{E}[G(\{\mathbf{z}_{g,i}\})] + 2\sqrt{2}\sqrt{\frac{\log(1/\delta)}{N_w}}, \tag{15}$$

which holds with a probability of at least $1 - \delta$ for every $\delta \in (0,1)$.

Next, for the function space $\Theta$, let

$$M_\theta = \sqrt{N_w} \sum_{g=1}^m \sum_{i=1}^{n_g} r_{g,i} \frac{\omega_g}{n_g} \left( -\ell(f_\theta(\mathbf{x}_{g,i}), y_{g,i}) \right), \ \forall \ \theta \in \Theta,$$

where $r_{g,i}$ are independent Rademacher random variables. For any $\theta_1, \theta_2 \in \Theta$, we define $d$ as

$$d^2(\theta_1, \theta_2) = N_w \sum_{g=1}^m \sum_{i=1}^{n_g} \frac{\omega_g^2}{n_g^2} \left( \ell(f_{\theta_1}(\mathbf{x}_{g,i}), y_{g,i}) - \ell(f_{\theta_2}(\mathbf{x}_{g,i}), y_{g,i}) \right)^2.$$

Now, we justify why this represents a random process with sub-Gaussian increments. In the definition of $M_\theta$, the Rademacher random variable $r_{g,i}$ introduces randomness from the sign of each term. $M_\theta$ can thus be understood as an empirical average over random sign flips. Additionally, $d^2(\theta_1, \theta_2)$ computes the squared difference in losses under $\theta_1$ and $\theta_2$. The squared term ensures that this quantity is non-negative and gives a measure of "distance" between the two functions in terms of their losses. Combining both observations, we have that the difference $M_{\theta_1} - M_{\theta_2}$ is a random process with increments characterized by the metric $d^2$. Moreover,

$$\mathbb{E}\left[ \exp\left( \lambda(M_{\theta_1} - M_{\theta_2}) \right) \right] \le \exp\left( \frac{\lambda^2}{2} d^2(\theta_1, \theta_2) \right), \ \forall \ \lambda \ge 0, \theta_1 \in \Theta, \theta_2 \in \Theta.$$

This result shows that the tail probabilities of the process $M_{\theta_1} - M_{\theta_2}$ decay at least as fast as those of a Gaussian process, justifying the sub-Gaussian property.

Next, we use Dudley's entropy integral inequality conditioned on the randomness of $\mathbf{z}_{g,i}$ to obtain

$$\mathbb{E}\left[ \sup_{\theta \in \Theta} M_\theta - M_{\bar{\theta}^w} | \{\mathbf{z}_{g,i}\} \right] \le 8\sqrt{2} \int_0^1 \sqrt{\log \mathcal{N}(\epsilon; \Theta; d)} d\epsilon. \tag{16}$$

Recall the definition of the covering number from Assumption B.2. Given that

$$\sqrt{N_w} \mathbb{E}[G(\{\mathbf{z}_{g,i}\})] \le 2\sqrt{N_w} \times \mathbb{E}\left[ \sup_{\theta \in \Theta} \sum_{g=1}^m \sum_{i=1}^{n_g} r_{g,i} \cdot \frac{\omega_g}{n_g} (\ell(f_{\bar{\theta}^w}(\mathbf{x}_{g,i}), y_{g,i}) - \ell(f_\theta(\mathbf{x}_{g,i}), y_{g,i})) \right],$$

Eq. (16) can be transformed to

$$\mathbb{E}[G(\{\mathbf{z}_{g,i}\})] \le \frac{16\sqrt{2}}{\sqrt{N_w}} \int_0^1 \sqrt{\log \mathcal{N}(\epsilon; \Theta; d)} d\epsilon. \tag{17}$$

Next, we consider the covering number of $\Theta$. We define

$$\mathbb{Q} = \sum_{g=1}^m N_w \frac{\omega_g^2}{n_g} \sum_{i=1}^{n_g} \frac{\delta_{\mathbf{x}_{g,i}}}{n_g}, \text{ where } \delta_{\mathbf{x}_{g,i}} \text{ represents a point mass at } \mathbf{x}_{g,i}.$$

Denote $N_\epsilon$ as $\mathcal{N}(\epsilon; \Theta; L^2(\mathbb{Q}))$. Let $\{\theta^{(1)}, \cdots, \theta^{(N_\epsilon)}\} \subseteq \Theta$ be an $\epsilon$-covering of $\Theta$ with respect to $L^2(\mathbb{Q})$. This implies that for any $\theta \in \Theta$, there exists $j \in \{1, \cdots, N_\epsilon\}$ such that

$$\|f_\theta - f_{\theta^{(j)}}\|_{L^2(\mathbb{Q})}^2 = \sum_{g=1}^{m} \sum_{i=1}^{n_g} N_\omega \frac{\omega_g^2}{n_g^2} \|f_\theta(\mathbf{x}_{g,i}) - f_{\theta^{(j)}}(\mathbf{x}_{g,i})\|^2 \le \epsilon^2.$$

In conclusion, from the previous steps, we have:

$$
\begin{aligned}
d^2(\theta, \theta^{(j)}) &= N_\omega \sum_{g=1}^{m} \sum_{i=1}^{n_g} \frac{\omega_g^2}{n_g^2} (\ell(f_\theta(\mathbf{x}_{g,i}), y_{g,i}) - l(f_{\theta^{(j)}}(\mathbf{x}_{g,i}), y_{g,i}))^2 \\
&\le C^2 N_\omega \sum_{g=1}^{m} \sum_{i=1}^{n_g} \frac{\omega_g^2}{n_g^2} \|f_\theta(\mathbf{x}_{g,i}) - f_{\theta^{(j)}}(\mathbf{x}_{g,i})\|^2 \\
&= C^2 \|f_\theta - f_{\theta^{(j)}}\|_{L^2(\mathbb{Q})}^2 \le C^2 \epsilon^2.
\end{aligned}
\tag{18}
$$

From the above, we have that

$$N(L_l \epsilon; \Theta; d) \le N_\epsilon \le \left(\frac{C_\Theta}{\epsilon}\right)^{v_\Theta} \Rightarrow \log N(\epsilon; \Theta; d) \le v_\Theta \left(\log(C_\Theta C) + \log\left(\frac{1}{\epsilon}\right)\right).$$

Applying the above to Eq. (17), we get

$$
\begin{aligned}
\mathbb{E}[G(\{\mathbf{z}_{g,i}\})] &\le \frac{16\sqrt{2}}{\sqrt{N_\omega}} \int_0^1 \sqrt{v_\Theta \left(\log(C_\Theta C) + \log\left(\frac{1}{\epsilon}\right)\right)} \, d\epsilon \\
&\le \frac{16\sqrt{2}\sqrt{v_\Theta \log(C_\Theta C)}}{\sqrt{N_\omega}} \left(1 + \int_0^1 \sqrt{\log\left(\frac{1}{\epsilon}\right)} \, d\epsilon\right).
\end{aligned}
\tag{19}
$$

Further applying the above to Eq. (15), we get

$$G(\{\mathbf{z}_{g,i}\}) \le \frac{16\sqrt{2}\sqrt{v_\Theta \log(C_\Theta C)}}{\sqrt{N_\omega}} \left(1 + \int_0^1 \sqrt{\log\left(\frac{1}{\epsilon}\right)} \, d\epsilon\right) + 2\sqrt{2}\sqrt{\frac{\log(1/\delta)}{N_w}}.$$

Finally, from Eq. (12), we can conclude that:

$$L_0(\hat{\theta}) - L_0(\bar{\theta}^w) \le C_\rho \left(\frac{16\sqrt{2}\sqrt{v_\Theta \log(C_\Theta C)}}{\sqrt{N_\omega}} \left(1 + \int_0^1 \sqrt{\log\left(\frac{1}{\epsilon}\right)} \, d\epsilon\right) + 2\sqrt{2}\sqrt{\frac{\log(1/\delta)}{N_w}}\right)^{\frac{1}{\rho}} + \text{dist}\left(\sum_{g=1}^{m} \omega_g P_g, P_0\right).$$

The proof of Theorem B.3 is thus finished. $\qquad\square$

## C    Illustration of Feature Similarity

In this section, we explore how our algorithm affects the learned features across groups. We use the protein graph dataset as an example. For every pair of groups $i$ and $j$, we compute a feature similarity score $s(i, j)$ between the last-layer features' covariance matrices. For group $i$, denote $X_i \in \mathbb{R}^{n_i \times d}$ as the feature vectors of $n_i$ samples with dimension $d$. Denote the covariance matrix as $X_i^\top X_i$. We use the rank-$r_i$ approximation to the covariance matrix $U_{i,r_i} D_{i,r_i} U_{i,r_i}^\top$, where $r_i$ is chosen to contain 99% of the singular values. Then, we measure the similarity as

$$s(i, j) = \frac{\|(U_{i,r_i} D_{i,r_i}^{1/2})^\top U_{j,r_j} D_{j,r_j}^{1/2}\|_F}{\|U_{i,r_i} D_{i,r_i}^{1/2}\|_F \|U_{j,r_j} D_{j,r_j}^{1/2}\|_F}.\tag{20}$$

In particular, higher values of $s(i, j)$ indicate greater similarity between $i$ and $j$.

We verify that after splitting the graphs based on sizes and average degrees, the feature similarity score indeed varies. In particular, for every pair of two groups, we train a model on their combined data set and measure the feature similarity score between them. At the same time, we compute the differences in average graph sizes and average degrees between the two groups. Figure 7a shows that the feature similarity score drops as the disparity between graph sizes grows. Figure 7b shows qualitatively similar results when we measure the disparity by average degrees.

Our method can improve the feature similarity between groups; this is shown in Figure 7c. Here we measure the averaged feature similarity score, over every pair of groups.

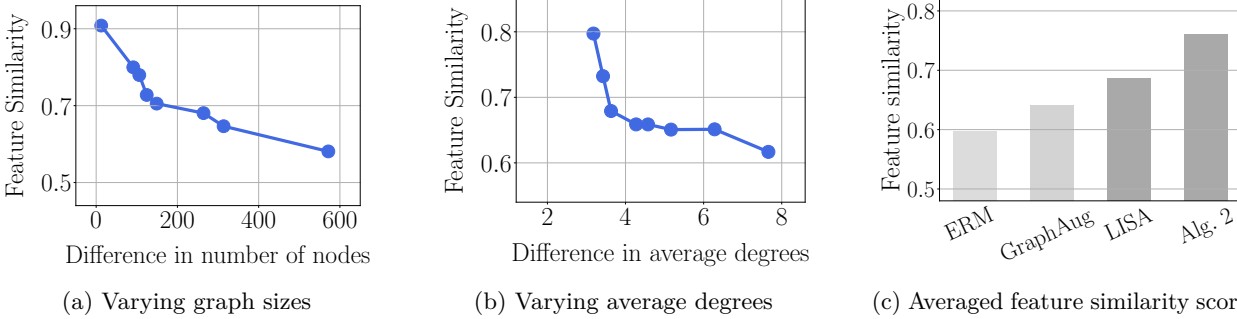

(a) Varying graph sizes   (b) Varying average degrees   (c) Averaged feature similarity score

Figure 7: We observe that groups with larger differences in graph size or average degree exhibit more distinct features in a trained model. Using augmentation can result in higher similarities between groups than not using it. In particular, our algorithm improves the similarity score of equation (20) between different groups. Here, ERM refers to empirical risk minimization.

