# OpenReview forum: "Learning Tree-Structured Composition of Data Augmentation"
_TMLR — Accepted by TMLR_

### Review · Reviewer_uCps · 2024-04-04

**Summary Of Contributions:**

The paper introduces a top-down recursive algorithm for efficiently searching within the space of tree-structured composition of transformations, where each node in the tree corresponds to one transformation. This algorithm significantly reduces the computational complexity from O(k^d) to O(2^{dk}), where k is the number of input transformations and d is the depth of the composition. This improvement is particularly notable as k increases beyond 2. The algorithm also address problems with data distributions with in heterogeneous subpopulations. It does so by partitioning the dataset into groups, finding one tree for each subpopulation, and then learning a weighted combination of these trees. This method not only improves computational efficiency but also enhances the performance by adapting to the variations within the data.

**Audience:**

Yes

**Claims And Evidence:**

Yes

**Requested Changes:**

Some suggestions -
* To ensure the generalizability of the proposed algorithm, it is essential to evaluate its performance across a wider array of datasets, particularly in domains significantly different from the ones tested.
* Providing comprehensive implementation details will help in better understanding.

**Strengths And Weaknesses:**

### Strengths

1. The introduction of a top-down recursive algorithm for finding tree-structured compositions of data augmentations is a significant strength. It offers a novel approach to handling the combinatorial explosion problem in data augmentation space, reducing the computational complexity from \(O(k^d)\) to \(O(2^{dk})\), where \(k\) is the number of transformations and \(d\) is the depth of the composition.

2. The extension of the algorithm to manage heterogeneous subpopulations by learning a forest of trees tailored to each subpopulation is particularly commendable. This approach not only enhances performance but also provides insights into the importance of specific augmentations for different data characteristics.

3. The comprehensive testing across several datasets, including a novel multi-label graph classification dataset derived from AlphaFold2, showcases the algorithm's effectiveness and efficiency. The demonstrated reduction in computation time and improvement in performance metrics are convincing arguments for the algorithm's utility.

### Weaknesses

1.  While the algorithm shows impressive results on the datasets tested, the paper could benefit from a broader evaluation across a more diverse set of domains. This would help ascertain the algorithm's generalizability and effectiveness in various practical applications beyond the ones explored.

2. Although the tree structures allow for some level of interpretation regarding the importance of different augmentations, the paper could further delve into the explainability aspect of the proposed method. Providing deeper insights into why certain transformations are more beneficial for specific subpopulations or tasks would enhance the paper's contribution.

3. While the paper introduces an innovative algorithm with demonstrated efficiency and effectiveness, an exploration of the sensitivity of the algorithm's performance to its hyperparameters (e.g., depth of the tree, number of transformations) could provide valuable insights for practical implementation. A detailed analysis could help users better understand the trade-offs involved and how to tune the algorithm for specific applications.

4. Providing more detailed implementation guidelines, code availability, and potentially a user-friendly interface or library could significantly enhance the paper's impact. This would allow a broader range of researchers and practitioners to apply the proposed method to their own problems without needing to navigate the complexities of algorithm implementation from scratch.

In conclusion, while the paper presents a significant advancement in the field of data augmentation with its innovative algorithm, addressing the above weaknesses could further solidify its contributions and facilitate wider adoption and adaptation of the proposed methods.

---

> ### Author Response · Authors · 2024-04-24
> **Responses to Reviewer uCps 1/2**
>
> We are very grateful to the reviewer for gently commenting on our work. We respond to the comments one by one below.
>
> **>> “The paper could benefit from a broader evaluation across a more diverse set of domains”**
>
> We emphasize that the core idea of this paper should be applicable to scenarios whenever a data augmentation scheme is used or needed. Recall that our current experiments already cover both supervised and self-supervised learning settings, spanning both graph data sets and image data sets. Below, we provide another set of experiments, adopting our algorithm to semi-supervised learning, where we use data augmentation to perform consistency regularization (in the sense of the work of Xie et al., NeurIPS’20).
>
> In more detail, we conduct semi-supervised learning on two data sets (CIFAR-10 and SVHN).
>
> - The former contains 1000 labeled examples and 49000 unlabeled examples.
> - The latter contains 1000 labeled examples and 64932 unlabeled examples
>
> We focus on the Wide-ResNet-28 architecture. We report some preliminary results below. On both datasets, by running Algorithm 1 to get the augmentation scheme, we could outperform UDA, which uses random sampling as the backbone scheme by the work of Xie et al., 2020.
>
> This experiment demonstrates the utility of applying our idea to construct augmentation schemes in semi-supervised learning. We will include this new setting in the updated manuscript.
>
> | Error rate (%)     | CIFAR-10     | SVHN       |
> | ----------------------- | ----------------- | ----------------- |
> | UDA (Xie et al., 2020) | 4.9 $\pm$ 0.1   | 2.7 ± 0.1     |
> | Algorithm 1 (This work) | **4.5** $\pm$ 0.1 | **2.2** $\pm$ 0.1 |
>
> Xie et al., Unsupervised data augmentation for consistency training, NeurIPS’20.
>
> **>> “The paper could further delve into the explainability aspect of the proposed method. Providing deeper insights into why certain transformations are more beneficial for specific subpopulations or tasks would enhance the paper's contribution.”**
>
> To address this question, recall that based on the tree structure, one can calculate a score for each $A_i$. This is simply by adding up the overall decrease in the validation loss, across all the nodes corresponding to $A_i$, for each $i = 1, 2, \dots, k$.
>
> As an example, in Figure 4, we are showing the augmentation tree, based on running Algorithm 2 on a graph classification data set. We can observe that
>
> -  on a group of small graphs with less than 200 nodes, randomly removing edges in the graphs has a score of 0.12, which is the highest relative to the score of randomly removing nodes of the graph (0.02).
>
> - on a group of large graphs with more than 600 nodes, generating a subgraph by random walks in the original graph has the highest score of 0.26, relative to the score of randomly removing edges (0.04) or nodes (0.03) in the graph.
>
> Our interpretation of this finding is that larger graphs can benefit from more significant perturbations, as compared to smaller graphs.
>
> To add another example, on the wildlife image classification dataset, we observe that
>
> - on a group of colored images, the Auto Contrast transformation has the largest score of 0.22, which modifies the color distribution of images, relative to the score of Solarize (0.11) and Color Enhancing (0.09).
>
> - on a group of black-and-white images, the Equalize transformation has the largest score of 0.64, which modifies the grayscale distribution of images. This is relative to the score of TranslateY (0.25) and Posterize (0.09).
>
> We have now added these results to Section 3.2 of the revised paper.

---

> ### Author Response · Authors · 2024-04-24
> **Responses to Reviewer uCps 2/2**
>
> **>> “An exploration of the sensitivity of the algorithm's performance to its hyperparameters (e.g., depth of the tree, number of transformations) could provide valuable insights for practical implementation.”**
>
> First, we summarize the hyperparameters required by the algorithm:
>
> - In Algorithm 1: We require the depth of the tree $d$, a list of probability values $H$ uniformly between 0 and 1.
>
> - In Algorithm 2: We need two the number of SGD steps $\alpha$, the learning rate $\eta$ for updating the weights.
>
> Below, we vary these parameters and report the validation performance. We vary one at a time and keep the others unchanged.
>
> |             | Graph classification |          |          | Image classification |          |          |
> | :----------------------: | :------------------: | :----------------: | :----------------: | :------------------: | :----------------: | :----------------: |
> |             |    Validation AUROC     |          |          |   Validation Macro-$F_1$   |          |          |
> |  Max tree depth $d$  |     2      |     3     |     4     |     2      |     3     |     4     |
> |             |  72.4 $\pm$ 0.1  |  74.2 $\pm$ 0.2  | **74.5** $\pm$ 0.4 |  61.6 $\pm$ 0.6  |  65.0 $\pm$ 0.3  | **65.3** $\pm$ 0.6 |
> | # of probabilities $H$  |     5      |     10     |     20     |     5      |     10     |     20     |
> |             |  72.9 $\pm$ 0.2  | **74.5** $\pm$ 0.4 | **74.5** $\pm$ 0.6 |  64.7 $\pm$ 0.3  | **65.3** $\pm$ 0.6 | **65.3** $\pm$ 0.4 |
> |  SGD Steps $\alpha$  |     25     |     50     |    100     |     25     |     50     |    100     |
> |             |  74.0 $\pm$ 0.2  | **74.5** $\pm$ 0.4 |  73.4 $\pm$ 0.8  |  64.8 $\pm$ 0.3  | **65.3** $\pm$ 0.6 |  64.2 $\pm$ 0.4  |
> |  Learning rate $\eta$  |     1.0     |    0.1     |    0.01    |     1.0     |    0.1     |    0.01    |
> |             |  71.0 $\pm$ 0.1  | **74.5** $\pm$ 0.4 |  72.4 $\pm$ 0.8  |  60.5 $\pm$ 0.7  | **65.3** $\pm$ 0.6 |  64.3 $\pm$ 0.5  |
>
> We summarize the above results as follows:
>
> - Setting depth $d$ as 4 and number of probabilities $H$ as 10 leads to the best results for both cases. Further increasing the depth and probability values brings no obvious performance gains.
> - Setting $\alpha$ as 50 and $\eta$ as 0.1 leads to the best results. We also observe that this holds for other datasets.
>
> Based on this study, we set these as the default in the experiments. We have now added the details in Section 4 of the revised paper.
>
> **>> “Providing more detailed implementation guidelines, code availability, and potentially a user-friendly interface or library could significantly enhance the paper's impact.”**
>
> We have set up a code repository for reproducing the experimental results in this link: https://anonymous.4open.science/r/Learning-Tree-Structured-Composition-of-Data-Augmentation
>
> The data set can also be accessed in a link: https://drive.google.com/file/d/1Bbl-urCGbnUNim6A0ddobI8MMxeiW9Bu/view?usp=sharing and then loaded into our code using train.py in the protein graph classification folder.
>
> We plan to release these after the review process.

---

### Review · Reviewer_BzR9 · 2024-04-20

**Summary Of Contributions:**

The paper proposes an algorithm for tree-based search for composition of data augmentations. It further proposes a weighted scheme for learning a model over sub-populations within the data, each with specific augmentation tree. The algorithm excels in runtime as compared to existing augmentation-search methods and improves (or is on par) with these in respect to supervised and self-supervised performance metrics. In the experimental section, the paper also introduces a new dataset for classification of protein graphs.

**Audience:**

Yes

**Broader Impact Concerns:**

No concerns

**Claims And Evidence:**

Yes

**Requested Changes:**

*1. Please clarify the augmentation procedure to address these concerns:*
* I am confused by the construction of the trees. If I understand these re binary trees (in the sense of every node having two children). However, in Figure 1, the first graph in subfigure 2 seems to have only single child at every step.
* This gets even more confusing when we consider "No transformation" or "Identity mapping". Are these included as transformations on its own in the list of possible transformations? Or how can these be introduced into the tree otherwise.
* Another confusing thing is the binary split probability and identity of the alternative. It seems to me from Algo 1, that at each step a single transformation is picked, tested and placed in the tree as a child with appropriate probability based on testing its performance over a validation set. How do I decide on its alternative? That is if augmentation $A_i$ is chosen with probability $p_i$ what happens with remaining probability $1-p_i$. What is the alternative to $A_i$? No transformation and stop growing the tree? Or Identity transformation so skip and go to next level? This would seem natural (and seems to be suggested in Figure 2) but in Figure 3 and 4 it seems the alternative is none of these.

*2. Please clarify the model learning description to clarify the below:*
* I gathered that the algorithm operates in two independent steps. First the augmentation trees are constructed for every sub-population, then a common model is learned by the weighted scheme. The trees for each sub-population may be different. Are the individual models of the subpopulations also different at this step or are they somehow synchronised through the updates of step 9. in Algo 1? (I do understand that Algo 2 learns a common model.)
* If they are not synchronized, I would be interested to know what is the effect of these models possibly diverging and then being forced to align through the second step. The augmentation trees are tuned for the model of Algo 1 and this may in the end be rather different than Algo 2. Please comment.
* Minor: the fact that the algo learns a single model over the sub-populations is correctly stated in the equations but is somewhat difficult to understand at first sight. Please state it clearly somewhere early in the text to help the reader to follow.

*3. Minor changes:*
* Section 2, first para: I believe the loss function should read $l: \mathcal{Y} \times \mathcal{Y} \to \mathbb{R}$. That is mapping from Y spaces to R, not from X and Y to R.
* Section 3.2, second para, second sentence: "Since labeling the protein ... " - please elaborate. I do not understand what does this mean.
* Figure 5: please add titles to the graphs (to recognize quickly what they contain without having to read the caption)

**Strengths And Weaknesses:**

**Strengths:** The paper is well motivated and presented. The proposed algorithms demonstrate superior performance as compared to existing methods. The appendix provides further mathematical details to strengthen the arguments.

**Weaknesses:**
* The discussion of the augmentation algorithms is somewhat confusing, especially in the treatment of "no transformation" and "identity mapping". See Requested Changes.
* I didn't find a link to the implementation of the algo.
  * This significantly reduces opportunities for further re-use and development of the algorithm.
  * Diminishes trust in the results as they cannot be re-produced
* I didn't find a link to access the newly created dataset.
  * This significantly reduces opportunities for further re-use and development of the algorithm.
  * Diminishes trust in the results as they cannot be re-produced

---

> ### Author Response · Authors · 2024-04-24
> **Responses to Reviewer BzR9**
>
> Thanks for reading through our manuscript in detail and providing very constructive feedback. Regarding the link to access the code and the data set, we have now put them up and they are available here:
>
> Code: https://anonymous.4open.science/r/Learning-Tree-Structured-Composition-of-Data-Augmentation
>
> One can directly load our newly constructed data set in the repository. First, download the dataset: https://drive.google.com/file/d/1Bbl-urCGbnUNim6A0ddobI8MMxeiW9Bu/view?usp=sharing.
>
> Then, place it under the protein function classification folder and run the `train.py` directly. We have included complete details about this in the online repository.
>
> We plan to make these public after the review process.
>
> We now respond to the remaining comments.
>
> **>> “I am confused by the construction of the trees… In Figure 1, the first graph in subfigure 2 seems to have only a single child at every step. Are "No transformation" or "Identity mapping" included.”**
>
> In the case of the first graph in subfigure 2, if a transformation is not applied, then the input is not changed and the sequence moves to the next node, just like the skip connection. We have now expanded this explanation in Figure 2 of the updated paper when we introduce the sequential augmentations.
>
> Besides that, we have updated all the other figures to make it clear when an identity mapping (i.e., $A(x) = x$) is used. Notice that under this design, the augmentations can take one step, two steps, three steps, etc., depending on when it stops. This adds flexibility to the augmentation scheme.
>
> As for the stopping criterion, if no transformation shows any benefit after adding it to the tree, which is the same as saying that inserting $A(x) = x$ will not introduce such benefit, then we will stop growing the tree.
>
> We have revised Algorithm 1 and the section to make this clear.
>
> **>> “If augmentation $A_i$ is chosen with probability $p_i$, what will happens with the remaining probability $1-p_i$?”**
>
> This depends on the other branch of the tree, as clarified above, by default, if there is no right node, then this means there is no transformation used, so we will just use $A(x) = x$. Then, no further transformation is applied after the identity mapping.
>
> To make this more clear, we have now revised all of the illustration figures to include the identity mapping in the tree.
>
> **>> “I gathered that the algorithm operates in two independent steps… Are the individual models of the subpopulations also different or are they synchronized through the updates of step 9. in Algo 1? If they are not synchronized…”**
>
> The models trained in the first step (in Algorithm 1) are not reused. The second step will train a new model by a weighted combination of losses on each subpopulation. On each subpopulation, the loss is computed by applying its corresponding augmentation found in the first step.
>
> The augmentation in each subpopulation can be quite different. This motivates us to design the weighted training step. This step will down-weight the loss of the subpopulation whose gradient does not align with the average gradient. For example, in the protein graph classification dataset, we split four groups of graphs, which have the average number of nodes as 150, 300, 500, and 1200, respectively. The graph sizes of the last group differ from the rest of the groups the most. Accordingly, we observe that the algorithm assigns the smallest weight (less than 0.1) to this group.
>
> We have validated that this weighted training algorithm is beneficial when combining different subpopulations. We compare it to the uniform weighted training, where each population applies its own augmentation and the loss on each subpopulation has the same weight. We observed that using Algorithm 2 can outperform the uniform weighted training by 1.0% and 1.6% on the protein graph classification and wildlife image classification data set, respectively.
>
> **>> “Minor: the fact that the algo learns a single model…”**
>
> We have now added a sentence before Algorithm 2 to emphasize this fact of the algorithm.
>
> **>> “Minor changes”**
>
> Thanks for noticing these issues. They have been fixed in the revised manuscript.
>
> - Re “Section 2, first para”: We have added a sentence to clarify this now.
> - Re “Section 3.2, second para”: We have revised this paragraph.
> - Re “Figure 5, subcaption”: We included a brief caption for each subfigure and did the same for the other figures in the main text.

---

### Review · Reviewer_aMpo · 2024-04-24

**Summary Of Contributions:**

The paper presents efficient algorithms for searching over a space of tree-structured compositions of transformations for the purpose of data augmentation. The algorithm has dramatically reduced worst-case runtime complexity, compared to existing approaches, for settings where there is a relatively large number of transformations to choose from. The paper includes an extension to address data distributions with heterogeneous subpopulations. The experiments, conducted on several graph and image datasets, provide support for the claimed advantages of the method, namely demonstrating that the proposed method can reduce computation substantially while achieving a small performance improvement. The paper discusses how the adoption of a tree structure can provide insights into which transformations are most effective for different types of data.  As a final contribution, the paper reports on a newly collected graph dataset.

**Audience:**

Yes

**Claims And Evidence:**

Yes

**Requested Changes:**

1.	The description of the proposed procedure (single tree) is brief (<1page). The presentation should be expanded considerably, with better justification and more in-depth discussion of the design decisions and their validity.
1a. What is the justification for training only one model at each step? What is the nature of the resultant approximation and what are the implications and ramifications? Why is it considered that this is a reasonable approximation? Please add more discussion on this point. Currently there is just a passing reference to another paper.
1b. Is there an understanding of how well or poorly the greedy search performs? What aspects of the problem influence this?

2.	Appendix C provides a generalization bound, but there is no discussion of this result in the main paper. Even the appendix does not clearly present the implications of the result with respect to the overall search procedure. Why are these results downplayed to this extent? They seem to embody a substantial effort (4 pages of theoretical work) and provide interesting insights. One sentence in passing in the main paper seems odd. Please add a paragraph in the main text that describes the nature of the result and the context.

3.	There are no citations in Appendix C. Please provide some discussion about related work – does this proof build on work in other papers? Is it related to results in the transfer learning literature? Is there use of proof techniques from other papers? The complete absence of citations is surprising.

4.	The approach for learning a forest of trees relies on identifying suitable groups and selecting the number of groups. Could the authors add text to explain how this is achieved? In the provided example the choice is “16 intervals of sizes and average degrees” but there is no explanation of how the number 16 is chosen or how the sizes and partitions are selected.

5.	Please add quantification of the variability of the proposed technique. This would be particularly useful in this setting to get a better understanding of the robustness of the greedy search (whether it is prone to find local minima). Please conduct appropriate statistical significance tests (e.g., Wilcoxon pair) to demonstrate that the reported differences are meaningful.

6.	Minor: I do not understand why Section 3.1.2 is included in the methodology (Our Proposed Algorithms). These aren’t examples that provide any further insight into how the algorithm works. They are performance experiments and belong in the experiments and results section. Please move them accordingly or justify what value they bring to the methodology section.

**Strengths And Weaknesses:**

Strengths

1.	The proposed algorithm is clearly explained and the idea of an efficient search over tree-structured compositions is sensible and effective.

2.	The experiments are extensive and thorough and demonstrate convincingly that the proposed method works well. Ablation studies are provided to demonstrate that the key components of the proposed procedure are all important in contributing to the performance improvement.

Weaknesses

1.	The proposed method is a greedy search procedure. In addition, the validation performance during the search is approximated, so that even the greedy search is over an approximate objective. There are no theoretical results characterizing the quality of the approximation or the ability of the overall algorithm to approximate the optimal solution. Theoretical results quantifying the generalization capability of the proposed bilevel optimization procedure are included, but these results, while commendable, do not address the greedy search and approximation issues.

2.	There is minimal in-depth experimental analysis to establish why the method works well.  The discussion in the main body of the paper does not go much beyond “Our method is better…” There is some investigation of the similarity of the selected features for different groups in Appendix B. Diagnostic experimental results would be a welcome inclusion in the paper.

3.	There is no reporting of variability; there are no statistical tests.

---

> ### Author Response · Authors · 2024-04-26
> **Responses to Reviewer aMpo 1/3**
>
> We thank the reviewer for carefully reading our paper and providing constructive feedback. We respond to each comment in detail below.
>
> \>> **“The proposed method is a greedy search procedure… There are no theoretical results characterizing the ability of the overall algorithm to approximate the optimal solution; Is there an understanding of how well or poorly the greedy search performs? What aspects of the problem influence this?”**
>
> Thanks for this insightful question. To the best of our knowledge, theoretical results in the data augmentation literature are scarce, despite the fact that there has been lots of progress in developing guarantees for training neural networks in the past few years.
>
> Part of the challenge is that the data augmentation training paradigm violates the independent sampling assumption that is typically required in the theoretical literature. For instance, suppose one would like to understand how applying a sequence of transformations, like rotation, cropping, etc, to an image, affects the downstream performance. This would require modeling such transformations within the data augmentation. There have been few developments in this direction, for instance, work by Chen et al. (2020) develops a group-theoretical framework modeling data augmentation. However, that work’s result only applies to a single transformation, and it does not work for the composition of multiple transformations. Another recent work by Shao et al. (2022) develops a PAC-learning framework for transformation-variant data sets. Their algorithm relies on a counting argument and is thus not very practical.
>
> Chen, S., Dobriban, E. and Lee, J.H. A group-theoretic framework for data augmentation. JMLR 2020.
>
> Shao, H., Montasser, O. and Blum, A. A theory of PAC learnability under transformation invariances. NeurIPS 2022.
>
> Having said the above, we note that data augmentation has generally been quite useful in practice. As hinted above, various transformations add invariance to the data set, which can in turn improve generalization, and this can be fleshed out in a high-dimensional linear regression setting (Wu et al. (2020)).
>
> Wu, S., Zhang, H., Valiant, G. and Ré, C. On the generalization effects of linear transformations in data augmentation. ICML 2020.
>
> Part of the contribution of this paper to this literature is showing that a simple greedy search on the tree compositions can achieve competitive results to existing methods. This is empirically verified in both supervised and self-supervised learning settings, across ten graph and image data sets, as our algorithm can outperform the best-performing baselines by up to 7.4% with 30%~50% less runtime. This is not known before, to the best of our knowledge.
>
> To further substantiate this claim, we compare our algorithm with the exhaustive search that enumerates every possible tree. Due to the computation limit, we apply the exhaustive search to find the best depth-2 tree (of three nodes). Then, we compare its performance with the depth-2 tree found by our algorithm. We notice that the performance of our algorithm is only 0.4% below the exhaustive search, averaged on the protein graph and wildlife image classification dataset.
>
> Notice that greedy recursive partitioning is a common method in tree-based methods. There are studies (albeit in a very different setting) on the approximation ratio of greedy search in constructing trees, e.g. Adler and Heeringa (2008) and Gupta et al. (2017). It is an interesting research direction to examine these results in the context of data augmentation. This likely requires the development of new techniques, which should be beyond the scope of our paper.
>
> Adler, M., & Heeringa, B. Approximating optimal binary decision trees. In Approximation, Randomization and Combinatorial Optimization. Algorithms and Techniques. 2008.
>
> Gupta, A., Nagarajan, V., & Ravi, R. Approximation algorithms for optimal decision trees and adaptive TSP problems. Mathematics of Operations Research 2017.

---

> ### Author Response · Authors · 2024-04-26
> **Responses to Reviewer aMpo 2/3**
>
> **>>** **“The validation performance during the search is approximated; What is the justification for training only one model at each step?... Why is it considered that this is a reasonable approximation?”**
>
> Recall that in each step of the top-down binary search procedure, we need to search for a choice over a list of transformations and probabilities. The naive search metric is the validation performance, which needs to train a model for each choice. Thus, we use the density matching technique to make the search in each step more efficient. This is a technique developed in prior works, e.g., Lim et al. (NeurIPS’19) and Hataya et al. (EVVC’20). The intuition of this technique is to measure how much the model predictions on the validation data set align with the predictions on the training data set. Thus, at each step, we train one model on the training data set and evaluate the performance of applying the augmentation on the validation data set.
>
> To better justify this, we compare this technique with the validation performance of training a model for each augmentation choice. First, we find that on both protein graph and wildlife image classification data sets, this technique finds the same tree as validation performance. Second, we compute the relative residual sum of squares error between the approximated and the true validation performances. We notice that the relative error is only 0.7% between the two metrics.
>
> **>> “There is minimal in-depth experimental analysis to establish why the method works well…There is some investigation of the similarity of the selected features for different groups. Diagnostic experimental results would be a welcome inclusion in the paper.”**
>
> One justification here is that the trees are a generalization of sequential augmentations, which have been the focus of prior work. Since we are searching inside this family of generalized structures, it is expected that our results should be on par (if not better) than the prior results. Again, we have to emphasize here that much of the results in this literature are empirical, so establishing why our (or any other) method works well would be an interesting question for further research.
>
> If the reviewer has a specific experiment in mind, we would be happy to include it in the paper.
>
> Please kindly note that we have already included ablation studies to justify both steps of the algorithm in the experiments.
>
> **>> “There is no reporting of variability; there are no statistical tests… Please conduct appropriate statistical significance tests (e.g., Wilcoxon pair) to demonstrate that the reported differences are meaningful.”**
>
> Thanks for the suggestion. We have now conducted the Wilcoxon signed-rank test on the performance of our algorithm and the baselines, as well as over the ten graph and image data sets in our experiments. The results show that our improvement over baselines is statistically significant.
>
> - Comparing our algorithm with the previous best augmentation search approach, the Wilcoxon test has a p-value of 0.0025.
> - Comparing our algorithm with RandAugment, the Wilcoxon test has a p-value of 0.0019
>
> Both of the above results show that the confidence of the null hypothesis is less than 1%, which suggests the improvement over the baseline is statistically significant. We will add the results to the experiments section to clarify.
>
> **>> “The description of the proposed procedure (single tree) is brief. The presentation should be expanded considerably.”**
>
> Thanks for the suggestion. We will expand Section 3.1 to incorporate the above discussion of the greedy binary search approach and the efficient hyper-parameter search technique.

---

> ### Author Response · Authors · 2024-04-26
> **Responses to Reviewer aMpo 3/3**
>
> **>> “Appendix C provides a generalization bound, but there is no discussion of this result in the main paper. Please add a paragraph in the main text that describes the nature of the result and the context. There are no citations in Appendix C. Please provide some discussion about related work.”**
>
> This result justifies the consistency of the algorithm, i.e., showing that when the number of samples goes to infinity, the gap between training and test errors will shrink to zero. This result uses techniques from Chen et al. (2022) (see also Hanneke et al. (2019)).
>
> Hanneke, S. and Kpotufe, S. On the value of target data in transfer learning. NeurIPS 2019.
>
> Chen, S., Crammer, K., He, H., Roth, D. and Su, W.J. Weighted Training for Cross-Task Learning. ICLR 2022.
>
> We will add the explanation and the citations to the revised manuscript.
>
> **>> “The approach for learning a forest of trees relies on identifying suitable groups and selecting the number of groups… How the sizes and partitions are selected?”**
>
> We split graphs of similar sizes into the same group. To decide the number of groups, we conduct a hyper-parameter search. We vary the number of groups between 4, 9, 16, and 25, which correspond to split graph sizes (and average degrees) into 2, 3, 4, and 5 intervals. We apply the algorithm to each setting and evaluate their validation performance. Splitting beyond 16 adds no obvious benefit, so we choose 16. We will clarify this in the experiments.
>
> **>> “I do not understand why Section 3.1.2 is included in the methodology... Please move them accordingly or justify what value they bring to the methodology section.”**
>
> Section 3.1.2 is like a set of (empirical) examples for illustrating the efficiency of the tree construction. We would like to give some examples here before getting into further details of the algorithm. In particular, here we conducted a search in the same setting as SimCLR (Chen et al. (2020)), and noted that our algorithm found a very similar tree to the one used in SimCLR. We are happy to revise this section if the reviewer has additional feedback about the purpose of this subsection.

---

### Author Response · Authors · 2024-04-29
**Summary of Responses and Revisions**

Thanks to the reviewers for giving feedback on our paper. Here is a summary of comments and our revisions to address the comments.

**Release code and data set.** We have set up a code repository to reproduce the experimental results and load our newly constructed data set. We plan to release these after the review process.

**Better explanation of the tree construction and the illustrative figures.** We have added a paragraph in Section 2 to describe the construction of a tree-based composition. Then, we expanded the description of our algorithms in both Section 3.1 and 3.2. We also modified all the illustrations in the paper to clarify the procedure regarding the role of identity mapping and when the search procedure is stopped.

**Empirical analysis including wider applications and statistical tests.** Following the reviewers’ suggestions, we added a new set of experiments and empirical analysis, listed as follows:

- We have now added results to justify our top-down binary search procedure in Section 3.1.
- We added another set of experiments of semi-supervised learning in Section 4.
- We’ve also added examples to interpret the tree-based compositions in Section 3.2.
- We expanded our discussion of hyper-parameters and added significance tests of the empirical results in Section 4.
- Furthermore, we added a paragraph to explain our theoretical results in Section 3.2.

We hope these responses and revisions help address the reviewers’ concerns. We are happy to discuss further issues.

**Revision update 06/23:** We are extremely grateful for all the review feedback put into our work!
- We have revised the paper thoroughly, including clarifying statements that are not sufficiently backed up, by either rewriting those sentences or adding references.
- We have reduced the notation inconsistency between \map, \to, etc.
- We added an experiment for applying our augmentation method to semi-supervised learning on medical images (Chest X-Rays). Compared to a previous method (UDA, "Unsupervised Data Augmentation for Consistency Training" by Xie et al. (NeurIPS'20)), we can use our approach to replace the RandAugment step in their work and, as a result, improve downstream performance.

We have also placed the experiment code in a public GitHub repo, which includes instructions for loading the graph data set.
Thanks!

---

### Decision · Action_Editor_6sPc · 2024-05-28

**Recommendation:** Accept with minor revision

**Comment:**

The problem of data augmentation is of significant importance across the spectrum of machine learning. The authors propose what I believe to be an intuitively clear and interesting approach for selecting sample transformations for creating a new training point instance. Although the approach of a top-down tree-based search is fairly simple, the added mixture of tree transforms idea for different data splits makes it rather interesting. The whole approach reduces the augmentation process complexity significantly.

There are extensive experimental results that show the utility of the method, but there are no theoretical results that can provably establish the improvement of the method over other approaches. I find the lack of analyses reasonable due to the difficulty of the problem.

The three reviews were mixed. Two reviewers were positive while one reviewer requested more information about the method, some analytical verifications and more data set evaluations. I agree with the latter that when there is no theoretical analysis, significant experimental evidence should be included. I personally find the results on Alphafold protein data very interesting, but to reach out to a broader audience, the authors should include 1-2 more examples from different applications domains.

Also, the revision of the paper seemed a bit rushed and I noticed inconsistencies in the new text wrt the old text. For example, the authors used \to in the original submission to indicate mappings, and then reverted to the word "to" in some of the red text. Another thing that I see as a minor weakness is that certain claims are made in a "nonscientific manner" (like "solving highly complex optimization problem"  - please mention complexity right there and also add a reference).

So, my only requirements are to add at least one more data set example and polish up the writing in the newly added text.

**Audience:**

I believe that a broad audience will be interested in this line of work since it has many practical applications (e.g., computational biology).

**Claims And Evidence:**

The problem of data augmentation is of significant importance across the spectrum of machine learning. The authors propose what I believe to be an intuitively clear and interesting approach for selecting sample transformations for creating a new training point instance. Although the approach of a top-down tree-based search is fairly simple, the added mixture of tree transforms idea for different data splits makes it rather interesting. The whole approach reduces the augmentation process complexity significantly.

There are extensive experimental results that show the utility of the method, but there are no theoretical results that can provably establish the improvement of the method over other approaches. I find the lack of analyses reasonable due to the difficulty of the problem.